# BrlR from *Pseudomonas aeruginosa* is a receptor for both cyclic di-GMP and pyocyanin

Feng Wang[1], Qing He[1], Jia Yin[1], Sujuan Xu[1], Wei Hu[1] & Lichuan Gu [1]

The virulence factor pyocyanin and the intracellular second messenger cyclic diguanylate monophosphate (c-di-GMP) play key roles in regulating biofilm formation and multi-drug efflux pump expression in *Pseudomonas aeruginosa*. However, the crosstalk between these two signaling pathways remains unclear. Here we show that BrlR (PA4878), previously identified as a c-di-GMP responsive transcriptional regulator, acts also as a receptor for pyocyanin. Crystal structures of free BrlR and c-di-GMP-bound BrlR reveal that the DNA-binding domain of BrlR contains two separate c-di-GMP binding sites, both of which are involved in promoting *brlR* expression. In addition, we identify a pyocyanin-binding site on the C-terminal multidrug-binding domain based on the structure of the BrlR-C domain in complex with a pyocyanin analog. Biochemical analysis indicates that pyocyanin enhances BrlR-DNA binding and *brlR* expression in a concentration-dependent manner.

[1] State Key Laboratory of Microbial Technology, Shandong University, Jinan, 250100 Shandong, China. These authors contributed equally: Feng Wang, Qing He. Correspondence and requests for materials should be addressed to W.H. (email: hw_1@sdu.edu.cn) or to L.G. (email: lcgu@sdu.edu.cn)

Pseudomonas aeruginosa is an opportunistic Gram-negative bacterium associated with various types of human infections, including cystic fibrosis (CF), burn wounds, and urinary tract infection[1,2]. These infections are difficult to eradicate after the biofilm forms[3]. Many studies have found that biofilms are not simply a diffusion barrier to antibiotics, but that bacteria within biofilms use distinct mechanisms to resist the action of antimicrobial agents[4,5]. The modulation of some genes and proteins makes the biofilm cells adapt quickly to harsh environmental conditions, such as the increased expression of multidrug efflux pumps, the decreased permeability of the cell, and the function of antibiotic-modifying enzymes[6,7]. In particular, the multidrug efflux pumps are at the intersection of antibiotic resistance and bacterial virulence[8,9]. Their over-expression is often induced by transcription factors that respond to small-molecule inducers[10–12]. Therefore, understanding the mechanism of biofilm-specific transcriptional activation of efflux pumps may facilitate the development of novel treatments for refractory P. aeruginosa infections.

Since the turn of the 21st century, the molecular mechanisms underlying bacterial biofilm formation have been extensively studied. The intracellular second messenger cyclic diguanylate monophosphate (c-di-GMP) has emerged as a major signal that inhibits the motile planktonic lifestyle and induces formation of sessile biofilm[13], and also regulates virulence, drug resistance and many other cellular functions[14]. c-di-GMP is synthesized by diguanylate cyclases (DGCs, with GGDEF domains), and degraded by specific phosphodiesterases (PDEs, with EAL and HD-GYP domains)[15]. The regulatory function of c-di-GMP is mainly achieved through a number of receptors, including degenerate metabolic domains (diguanylate cyclases and PDEs)[16,17], PilZ domains (Pfam domain PF07238)[18,19], AAA-type ATPase domains[20], transcription factors[21–24], MshEN domains (PRK11519 super family)[25,26], and riboswitches[27]. In addition to the centrally important messenger c-di-GMP, pyocyanin (5-N-methyl-1-hydroxyphenazine) has also been recognized as an important signal molecule for biofilm development in P. aeruginosa, which is synthesized from chorismate by two phzABCDEFG operons, as well as by phzM and phzS[28,29]. Pyocyanin facilitates biofilm formation by promoting extracellular DNA release[30–32], and its degradation by the demethylase PodA disrupts P. aeruginosa biofilms[33]. Similarly to c-di-GMP, pyocyanin enhances the resistance of P. aeruginosa to many antimicrobial reagents by up-regulating the expression of multidrug efflux pumps[34,35]. These efflux pumps may export pyocyanin, pyocyanin-related molecules, and multiple antibiotics, thus resulting in increased pathogenic infections and efflux-based drug resistance[2,36,37]. The overlap between c-di-GMP and pyocyanin's functions suggests the possibility of crosstalk between their regulated signaling pathways. Indeed, it is reported that FliA modulates the production of pyocyanin and other phenazine pigments by a c-di-GMP dependent mechanism in P. aeruginosa[38], and the binding of pyocyanin to RmcA activates its PDE activity on c-di-GMP in strain PA14[39]. Moreover, identification of more intracellular receptors for pyocyanin would represent a significant step toward understanding the mechanism of the pyocyanin signaling pathway.

BrlR, a member of the MerR family of multidrug efflux pump activators, consists of a structurally conserved N-terminal DNA-binding domain and a C-terminal GyrI-like binding domain. BrlR mediates the antibiotic resistance of P. aeruginosa by activating the expression of multidrug efflux pumps and repressing PhoPQ activation[40–42]. BrlR has been found to be a c-di-GMP-responsive transcriptional regulator that mediates the correlation between biofilm formation and antibiotic tolerance of P. aeruginosa[5,43]. Intriguingly, BrlR does not contain any sequence suggesting a secondary structure similar to known c-di-GMP binding sites, and how c-di-GMP enhances BrlR's transcriptional activity is still unknown. The C-terminal domain of BrlR contains a conserved multidrug-binding pocket resembling the GyrI-like binding domain in BmrR that recognizes a diverse set of ligands[11,44,45]. One common feature of the transcriptional regulators in MerR family is that they are able to regulate their own expression upon binding the specific ligands[11]. Pyocyanin has been shown to cause the up-regulation of brlR gene[35], which makes it possible to be a potential inducer for BrlR. However, whether the C-terminal domain of BrlR contains a c-di-GMP binding site, and whether BrlR acts as a pyocyanin receptor to activate the expression of multidrug efflux pumps, remain to be determined.

In this study, we solve the crystal structures of apo-BrlR and BrlR in complex with c-di-GMP. The structures show that BrlR has two separate c-di-GMP binding sites in its DNA-binding domain. The HTH motif of the DNA-binding domain is self-blocked by the multidrug-binding domain and may result in weaker BrlR-DNA binding. The crystal structure of the BrlR-C domain in complex with a pyocyanin analog (3-amino-2-phenazino, 3A2P) shows that BrlR contains a rigid binding pocket for pyocyanin. Structural comparison indicates that the ligand occupies the interface of the two terminal domains and significantly relocates the DNA-binding domain of BrlR. These structural studies combined with biochemical data support that BrlR is a receptor for both c-di-GMP and pyocyanin, thus establishing a link between biofilm formation and multidrug efflux pumps expression.

## Results

**Overall structure of apo-BrlR and BrlR–c-di-GMP.** The structure of c-di-GMP-bound BrlR has been solved to 2.5 Å resolution by using single-wavelength anomalous dispersion (SAD) method (detailed data are in Table 1). Subsequently, the structure of apo-BrlR is determined by molecular replacement (MR) using a protein monomer of BrlR–c-di-GMP complex as the search model. The final model of apo-BrlR is refined to 3.1 Å resolution and contains four BrlR molecules in the asymmetric unit (Fig. 1a). Two full-length monomers (A and D) can be traced, and the other two (B and C) have the same interruptions (residues 32–36 and residues 138–143). The monomer can be divided into three domains (Fig. 1b): the N-terminal DNA-binding domain (residues 1–75), which adopts a winged helix-turn-helix fold; the long central α-helical linker (residues 76–119), which connects two terminal domains; and the C-terminal multidrug-binding domain (residues 120–270). The topology of the DNA-binding domain is α1-α2-β1-β2-α3-α4, which contains four α-helices and an antiparallel β-sheet, and is largely preserved in the MerR-like superfamily[45–47]. The structure of the multidrug-binding domain, a distorted eight-stranded β-barrel surrounded by two α-helices, is also very similar to the previously reported BmrR structures[44]. Four BrlR subunits assemble into a stable homotetramer which is composed of two cross-stacked dimers (AD, BC) (Fig. 1a, c). Each dimer is stabilized by an antiparallel coiled-coil formed by two longer, central α-helix linkers (Fig. 1a).

The structure of the BrlR–c-di-GMP complex belongs to space group I4. Each asymmetric unit contains two BrlR monomers and four c-di-GMP molecules (Fig. 1d). The topology is almost the same as that of the apo structure, which contains 7 α-helices and 10 β-strands in total. The monomer of the BrlR–c-di-GMP complex contains two c-di-GMP binding sites in different regions on the DNA-binding domain. Crystallographic symmetry analysis indicates that the BrlR-c-di-GMP complex also forms a stable homotetramer, which is made up of two dimers as the final model of apo-BrlR (AA′ and BB′, Fig. 1e). Tetramerization of the

**Table 1 Crystallographic data collection and refinement statistics**

| | Apo-BrlR | BrlR/c-di-GMP complex | BrlR-C/pyocyanin-analog complex | I3C-derivative BrlR/c-di-GMP complex |
|---|---|---|---|---|
| *Data collection* | | | | |
| Wavelength (Å) | 0.9792 | 0.9792 | 0.9792 | 1.54 |
| Space group | P65 | I4 | P21 | I4 |
| *Cell dimensions* | | | | |
| $a, b, c$ (Å) | $a = 111.8, b = 111.8, c = 260.2$ | $a = 135.9, b = 135.9, c = 94.6$ | $a = 39.7, b = 103.6, c = 39.8$ | $a = 136.1, b = 136.1, c = 95.4$ |
| $\alpha, \beta, \gamma$ (°) | $\alpha = \beta = 90.0, \gamma = 120.0$ | $\alpha = \beta = \gamma = 90.0$ | $\alpha = \gamma = 90.0, \beta = 119.0$ | $\alpha = \beta = \gamma = 90.0$ |
| Resolution (Å) | 50–3.10 (3.21–3.10) | 50–2.50 (2.59–2.50) | 50–1.40 (1.45–1.40) | 50–2.80 (2.90–2.80) |
| Redundancy | 10.9 (11.4) | 7.3 (7.5) | 6.6 (6.5) | 7.5 (7.5) |
| Completeness (%) | 99.7 (100) | 99.9 (100) | 98.5 (99.7) | 100 (100) |
| $I/\sigma(I)$ | 30.4 (7.5) | 27.2 (6.2) | 40.0 (10.1) | 25.9 (5.1) |
| $R_{merge}$ (%) | 8.1(50.2) | 7.1(49.3) | 8.4 (24.3) | 7.7(46.8) |
| *Refinement* | | | | |
| Resolution (Å) | 38.84–3.11 (3.21–3.11) | 48.06–2.50 (2.58–2.49) | 34.79–1.40 (1.42–1.40) | |
| Rwork/Rfree | 26.70/32.20 | 21.40/26.30 | 14.40/18.20 | |
| No. of reflections | 32807 | 29936 | 54292 | |
| *No. residues or atoms* | | | | |
| Protein | 8683 | 4424 | 2388 | |
| Ligand | 0 | 260 | 46 | |
| Water | 0 | 169 | 520 | |
| B-factors (Å²) | 118.0 | 61.0 | 23.0 | |
| Protein | 118.0 | 60.5 | 20.7 | |
| Ligand | | 98.3 | 30.5 | |
| Water | | 60.4 | 34.4 | |
| *R.m.s. deviations* | | | | |
| Bonds (Å) | 0.004 | 0.006 | 0.005 | |
| Angles (°) | 0.853 | 1.025 | 0.858 | |
| *Ramachandran plot (%)* | | | | |
| Favored | 93.8 | 96.8 | 96.6 | |
| Allowed | 5.9 | 3.2 | 3.4 | |
| Outlier | 0.4 | 0 | 0 | |

Data for high resolution shells is shown in parenthesis where applicable

BrlR and BrlR–c-di-GMP complex was confirmed by the results of the gel-filtration as well as the analytical ultracentrifugation assays (Fig. 1f and Supplementary Fig. 1). Recently, the structure of BrlR in complex with c-di-GMP was also reported by Raju and Sharma[48]. The reported structure and c-di-GMP-bound BrlR structure are quite similar, with an RMSD of ~ 0.2 Å (Supplementary Fig. 2). The finding also suggests that a BrlR tetramer with each monomer contains two identical c-di-GMP binding sites.

**Unique polymeric form of BrlR.** Members of the MerR family assemble into dimers and subsequently to exert their regulatory function by combining and modulating the pseudo-palindromic *mer* operator sequence[49]. Unlike other MerR-like proteins, BrlR forms a distinctive dimer of dimers. A DALI search for globally similar proteins revealed that BrlR is significantly different in structure from other MerR family proteins[50]. The highest Z-score for BrlR homolog BmrR (PDB entry 3D70 and sequence identity of 20%) is 12.6, with an r.m.s. deviation is 18.5 Å for 219 equivalent Cα positions, thus indicating that BrlR is in a strikingly different conformation.

Structural comparison showed that the three individual domains are quite conserved in these BrlR homologs (Supplementary Fig. 3), while the relative spatial positions of the DNA-binding domain and the multidrug-binding domain are not (Fig. 1g). A detailed structural examination showed that the C-terminal multidrug-binding domain (named the BrlR-C domain) rotates around the central α-helix linker ~155°, thus resulting in a higher r.m.s. deviation value in the structural comparison. The head-to-tail interaction of BrlR in a tetramer occurs only

between two adjacent dimers (AC vs. BD). Helix α1 and the subsequent turn of the HTH motif occupy the groove of another BrlR-C domain in the BrlR tetramer, with helix α3 interacting with helix α7′ of the BrlR-C domain (Fig. 1h). The unusual interface of BrlR tetramer causes most of its DNA-binding region blocked by the multidrug-binding domains from their tetrameric partners, thus indicating a lower DNA-binding ability. To our knowledge, the tetrameric BrlR reported here provides the first example of a self-blocked MerR-like transcriptional regulator.

**Two c-di-GMP binding sites in the DNA-binding domain of BrlR.** The most striking feature of the BrlR–c-di-GMP complex is the c-di-GMP-binding mode. The DNA-binding domain contains two separate c-di-GMP-binding sites, and shares a mutually intercalated c-di-GMP dimer in a stacking conformation with a crystallographic symmetry related binding site (Fig. 2a–c). Thus, the overall structure reveals a receptor–ligand complex with a 1:2 stoichiometry. The interactions stabilizing the c-di-GMPs are different for the two binding sites (Fig. 2d, e, and details are described in Supplementary Note 1). The structure shows that the 2′-hydroxyl group of the c-di-GMP molecule in each site points toward the solvent and is not involved in the binding to BrlR (Fig. 2d, e). This structural feature allows us to perform fluorescence polarization (FP)-based binding assays[19,23,24] by using a fluoresceinated c-di-GMP analog, 2′-Fluo-AHC-c-di-GMP (F-c-di-GMP), containing a fluorophore attached to one of the 2′-ribose hydroxyls (Fig. 2f). The result showed that BrlR binds F-c-di-GMP with a $K_D$ of $7.3 \pm 0.5$ μM (Fig. 2g), which is in agreement with the $K_D$ value obtained

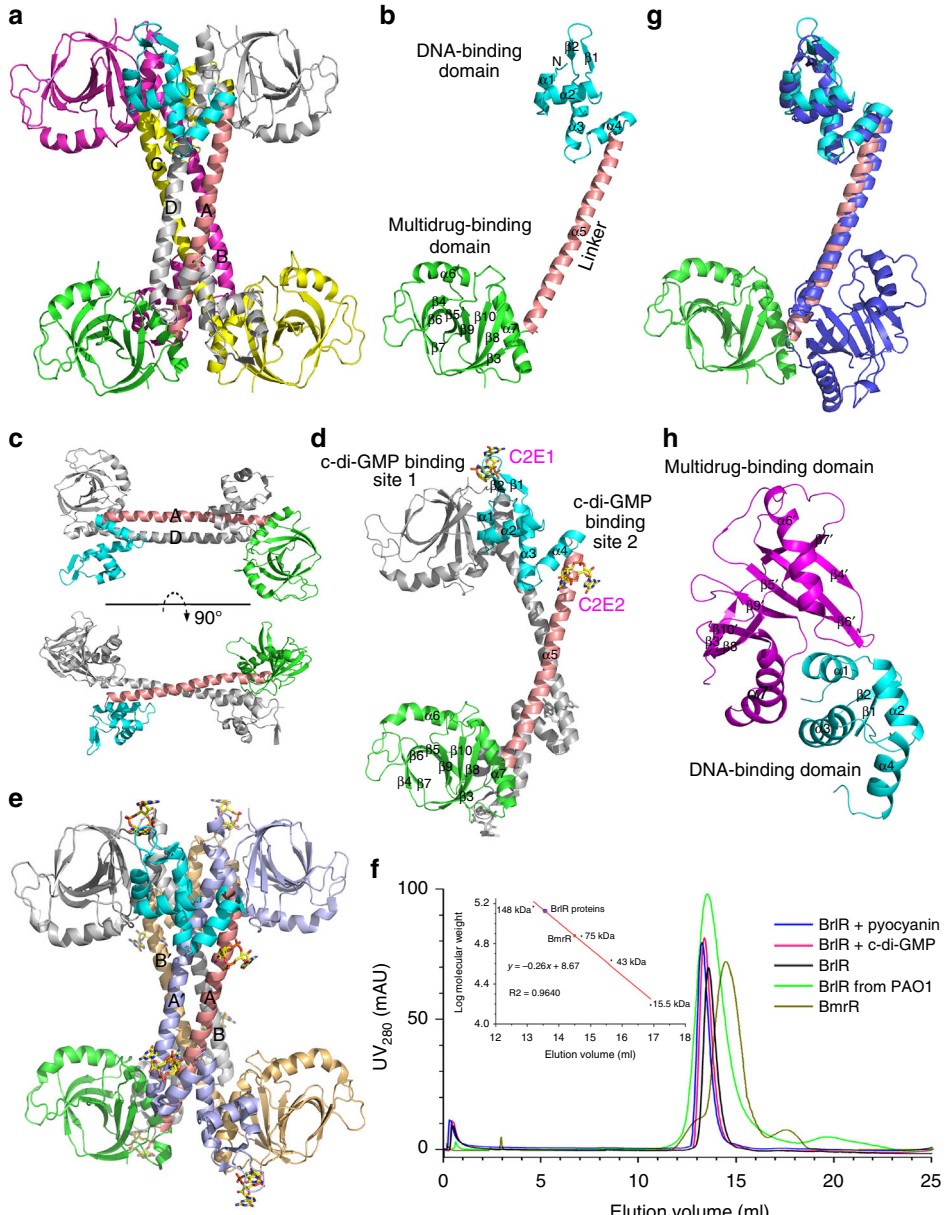

**Fig. 1** Crystal structures of BrlR and the BrlR–c-di-GMP complex. **a** Overall structure of the apo-BrlR. Protein monomers (cartoon) are indicated in different colors, the DNA-binding domain, a-helical linker and multidrug-binding domain of one monomer are shown in cyan, salmon, and green, respectively. **b** BrlR monomer. Three domains are colored as the corresponding monomer in **a**. **c** The stacked BrlR dimer from apo-BrlR, two dimers form a homotetramer. **d** Two monomers of BrlR–c-di-GMP complex in an asymmetric unit. One monomer is shown as in **b**, one is shown in gray. Sticks diagram depicting two individual c-di-GMPs which are labeled C2E1 and C2E2, respectively. **e** The homotetramer form of BrlR–c-di-GMP complex (cartoon), A and B monomers are shown as in **d**. A' and B' are indicated in blue and wheat, respectively. **f** Size-exclusion chromatography of BrlR, BrlR with c-di-GMP and pyocyanin using a Superdex 200 column. The calibration curve was generated using the standard proteins. **g** Structural superimpositions of the BrlR monomer and the BmrR monomer (PDB code: 1R8E, color in blue). BrlR is colored to match the cartoon in Fig. 1b. **h** The unique interface of one multidrug-binding domain of a stacked BrlR dimer (magenta) and an adjacent DNA-binding domain of the other dimer within the same tetramer (cyan). Secondary-structure elements referred to in the text are labeled

by Chambers et al[5]., and demonstrates that BrlR binds c-di-GMP efficiently. Notably, the binding of F-c-di-GMP to BrlR could be outcompeted specifically by c-di-GMP (Supplementary Fig. 4). Consistent with the structure data, the stoichiometry experiment revealed a 1.1:2 ratio of BrlR:c-di-GMP binding (Fig. 2h).

The FP binding analyses of the mutants involved in two separate c-di-GMP binding sites show that both the multiple-residue mutants of BrlR (R31A/D35A/Y40A/Y270A named C2E mut1) and the BrlR mutant (R67A/R86A named C2E mut2) are defective in c-di-GMP binding (Fig. 2g and Supplementary Note 2). Sequence alignment analysis shows that the residues in contact with c-di-GMPs are not conserved in MerR family proteins (Supplementary Fig. 5a), therefore, the effects of these single residues on c-di-GMP binding were also tested (Supplementary Fig. 6a, b). The results suggest that both c-di-GMP binding sites of BrlR contribute to c-di-GMP binding (Supplementary Note 3).

The FP assays show that, in addition to F-c-di-GMP, BrlR binds 2'-fluo-AHC-c-di-AMP (F-c-di-AMP) with a $K_D$ of 12.9 ±

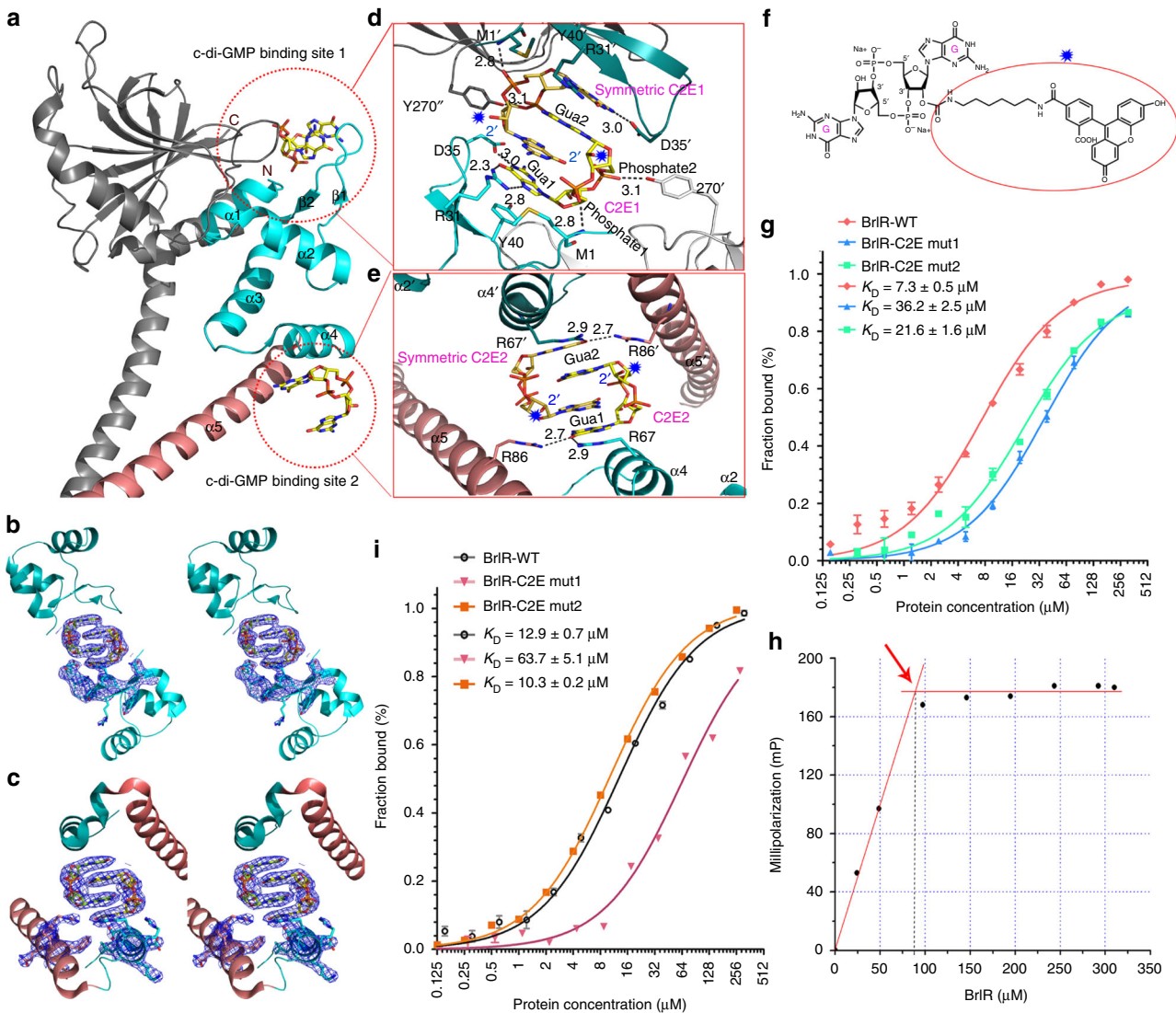

**Fig. 2** Each BrlR monomer contains two separate c-di-GMP binding sites. **a** Close view of the BrlR–c-di-GMP complex. Stick diagram depicting two stacked c-di-GMP binding sites in yellow. Two c-di-GMP binding sites are indicated by red dotted circles. **b, c** Well-defined 2Fo-Fc electron density map of the mutually intercalated c-di-GMP dimer and some surrounding amino acids countered at 1.0σ level along a stereoscopic view of the c-di-GMP binding site 1 (**b**) and the c-di-GMP binding site 2 (**c**). **d, e** An enlarged view of the amino acid residues in contact with the c-di-GMP dimer in c-di-GMP binding site 1 (**d**) and c-di-GMP binding site 2 (**e**). **f** Chemical structure of F-c-di-GMP; the fluorophore is indicated with a blue star. The blue stars in **d** and **e** indicate that the attached fluorophore has no interactions with the protein. **g** Binding of BrlR and BrlR mutants to F-c-di-GMP on c-di-GMP binding site, n = 3. The binding isotherms were fit to deduce the binding affinities. **h** The stoichiometry experiment involving BrlR binding with c-di-GMP. The inflection point occurs at a concentration of ~88 μM BrlR protein (red arrow), indicating a shift from high-affinity binding to no binding. Thus, the binding stoichiometry of c-di-GMP to BrlR was calculated as the initial concentration of c-di-GMP (160 μM) to the BrlR concentration (88 μM) at the inflection point, revealing a 1.1:2 ratio of BrlR:c-di-GMP binding. **i** FP analyses of F-c-di-AMP binding to BrlR, n = 3

0.7 μM (Fig. 2i). Intriguingly, the stoichiometry experiment yielded a 1:1 ratio for BrlR:c-di-AMP binding (Supplementary Fig. 7). Structural analysis illustrates that a c-di-AMP placed in the first c-di-GMP binding site interacts with BrlR in the same way as the c-di-GMP does (Supplementary Fig. 8a), while a c-di-AMP placed in the second c-di-GMP binding site is not compatible with the surrounding amino acid residues (Supplementary Fig. 8b). Moreover, only C2E mut1 but not C2E mut2 is defective in c-di-AMP binding (Fig. 2i). It suggests that BrlR binds to c-di-AMP in the first c-di-GMP binding site.

**BrlR undergoes conformational changes after c-di-GMP binding.** BrlR has been found to be a tetramer in solution,

regardless of whether c-di-GMP binds (Fig. 1f and Supplementary Fig. 1). Superposition of the apo-BrlR structure and the BrlR–c-di-GMP structure revealed some conformational changes. Compared with the apo structure, the HTH motif and the 'wing' loop undergo a movement in c-di-GMP-bound BrlR. The coiled-coil linker gradually bends along the helical turn, thus resulting in a rigid body movement of the BrlR-C domain (Fig. 3a). A structural comparison of the monomers indicated that the c-di-GMP binding changes the relative positions of the two terminal domains, which largely move as a rigid body (Fig. 3b). Like other MerR family proteins, BrlR has a flexible coiled-coil linker[49]. Crystal packing analyses showed that the long coiled-coils linker has no interaction with the protomers of adjacent tetramers, and the direct interactions

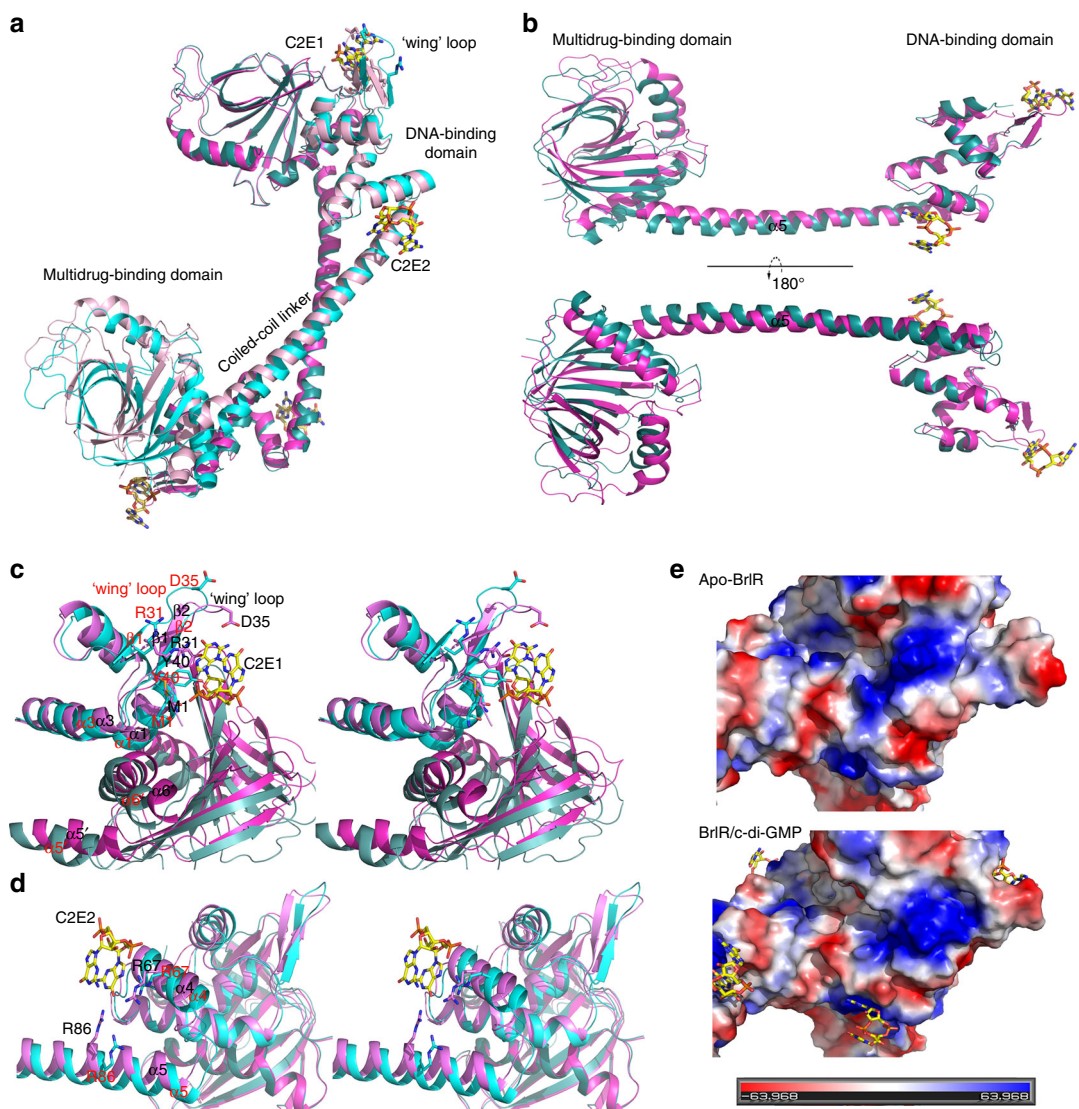

**Fig. 3** Detailed views of the c-di-GMP-induced domain movement and assembly of two distant c-di-GMP binding sites. **a** Structural superimpositions of the apo-BrlR (cyan and light cyan) and the c-di-GMP-bound BrlR (magenta and light magenta). **b** Superimposition of BrlR and c-di-GMP-bound BrlR in a monomer shows that the central helical linker of BrlR is flexible and two terminal domains undergo a clear rigid body movement after c-di-GMPs binding. **c** A close-up stereoview of the c-di-GMP binding site 1, the residues in contact with C2E1 and around the unique interface of BrlR homotetramer and undergo substantial repositioning after c-di-GMP binding. **d** A close-up stereoview of c-di-GMP binding site 2, the side chains of the residues in contact with C2E2 and the α-helices to which they belong, which twist significantly after c-di-GMP binding. **e** The DNA binding domain are shown as surface representations and colored according to their "in vacuum" electrostatics (red for negatively charged regions, and blue for positively charged regions, Pymol). Apo-BrlR is on the upper layer, and c-di-GMP-bound BrlR is on the lower layer. Secondary-structure elements and residues referred to in the text are labeled in Apo-BrlR (red) and the c-di-GMP-bound BrlR (black)

between the neighboring tetramers in the apo-BrlR structure appear more extensive than those in the BrlR–c-di-GMP structure (Supplementary Fig. 9a, b). This finding suggested that the conformational changes of BrlR are unlikely due to different crystal packing. To understand how c-di-GMP enhances BrlR-DNA binding, we further analyzed the conformational changes of the two c-di-GMP binding sites of BrlR in detail (Fig. 3c, d and Supplementary Note 4). Overall, the c-di-GMP-bound BrlR has a larger positively charged surface than apo-BrlR on the DNA-binding domain (Fig. 3e), and the simultaneous occupation of the two c-di-GMP binding sites twists and bends the DNA-binding domain and the coiled-coil linker (Fig. 3a, b). Consequently, the spacing and orientation between the two DNA-binding domains is altered, thus strengthening BrlR-DNA binding.

**Both c-di-GMP sites of BrlR are functional.** Electrophoretic mobility shift assays (EMSAs) were performed to detect the interaction between BrlR and the *brlR* promoter sequence (Supplementary Table 1). The EMSA results showed that BrlR specifically binds to P*blrR* (Fig. 4a), which is consistent with previous reports[5,43]. The addition of increasing concentrations of c-di-GMP or c-di-AMP significantly promoted the formation of BrlR-DNA complex (Fig. 4b). To further evaluate the effect of c-di-GMP on the BrlR-DNA interaction, we first performed EMSA for three c-di-GMP binding BrlR mutants with its target promoters P*brlR*, P*mexA*, and P*mexE* (in Supplementary Table 1). The results show that the C2E mut1 or C2E mut2 have low DNA-binding capabilities, suggesting that these mutations might change the surface charge of the HTH motif in BrlR. The DNA-binding capabilities of these mutants increased in the presence

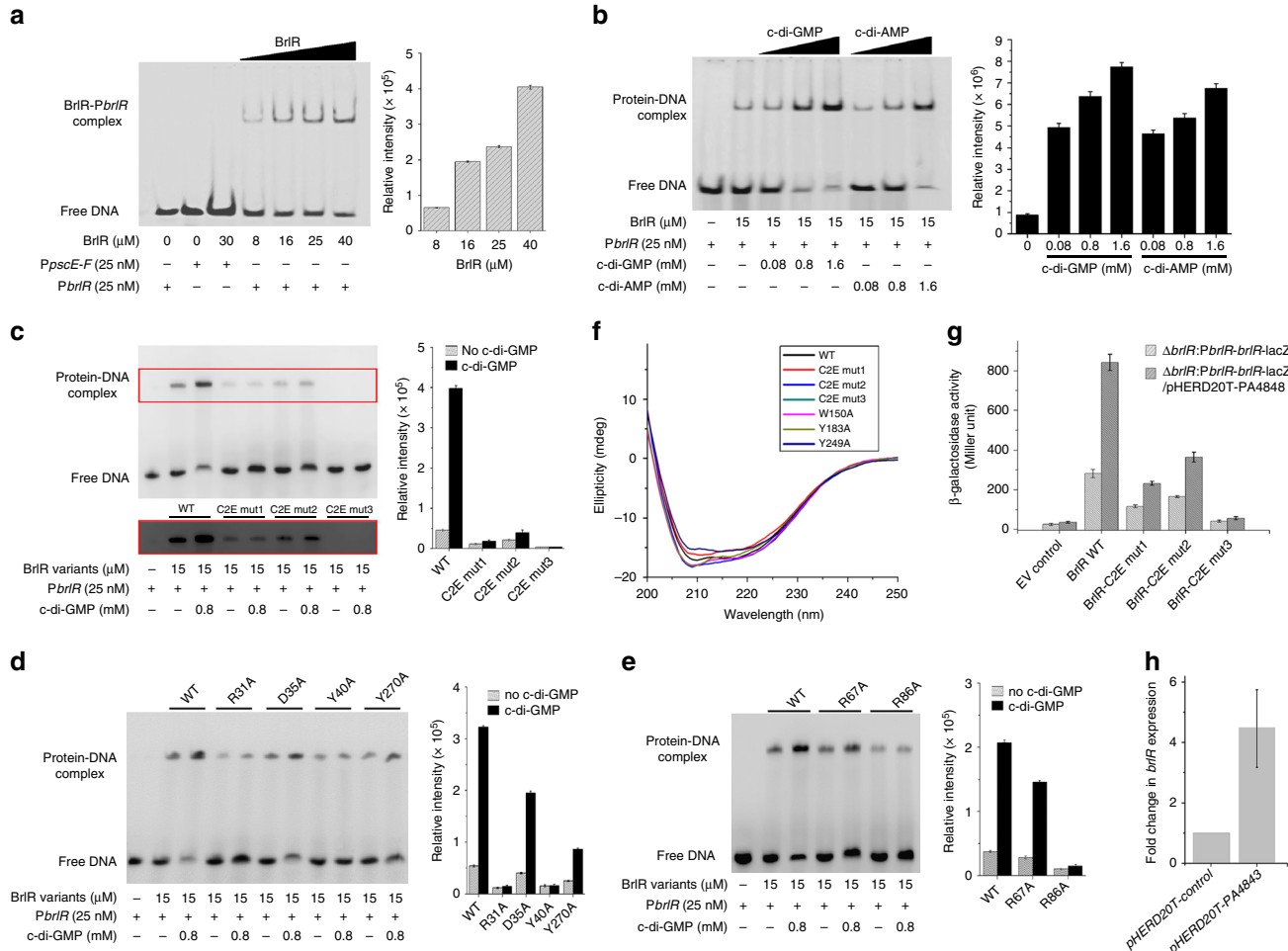

**Fig. 4** Both c-di-GMP binding sites of BrlR affect BrlR-DNA binding. **a** Electrophoretic mobility shift assays (EMSAs) demonstrating specificity of BrlR binding. A 36-bp FAM-labeled P*brlR* or P*pscE-F* DNA (0.375 pmol) was incubated with increasing amounts of BrlR protein (the concentrations are noted in the panel). The results were quantified by band densitometry (right). Error bars are s.d. for triplicate experiments. **b** BrlR-DNA gel mobility shift assays using FAM-labeled P*brlR* DNA (0.375 pmol) in the absence or presence of increasing concentrations of c-di-GMP and c-di-AMP. A total of 225 pmol of purified BrlR was used. The BrlR-DNA complexes were quantified by band densitometry (right). **c–e** BrlR-DNA gel mobility shift assays using BrlR mutants and P*brlR* DNA in the absence and presence c-di-GMP. The protein-DNA complexes within the top red rectangle are overexposed. **c** The mutants for two c-di-GMP binding sites. **d** The single point mutants of the first c-di-GMP binding site. **e** The single point mutants of the second c-di-GMP binding site. The BrlR-DNA complex was quantified by band densitometry on the right side of the gel. **f** Far-UV CD spectra (200–250 nm) were obtained for the wild-type BrlR and its related mutants, which were collected at ~20 μM at 25 °C on a Jasco J-810 spectropolarimeter. **g** Use of plasmid-borne *lacZ* transcriptional fusions to investigate the effect of BrlR protein on its own promoter at low and high c-di-GMP levels in *P. aeruginosa*. The promoter activities of *brlR* were tested in Δ*brlR* strains, and plasmid pHERD20T-PA4843 and pHERD20T were used to regulate c-di-GMP concentrations. An empty *lacZ* vector was used as a negative control. Error bars, s.d., obtained from triplicate experiments, $n = 3$. **h** The relative transcriptional changes in *brlR* are presented for *P. aeruginosa* PAO1 harboring pHERD20T-PA4843 and pHERD20T. Transcription levels of 16 S RNA were used as controls. Error bars, s.d., obtained from triplicate experiments, $n = 3$

of c-di-GMP. A double mutant of both c-di-GMP binding sites (R31A/D35A/Y40A/R67A/R86A/Y270A named C2E mut3) almost completely lost its DNA-binding abilities, and the addition of c-di-GMP did not have any effect (Fig. 4c; Supplementary Fig. 10a, b; and Supplementary Note 3). We next tested whether each residue involved in c-di-GMP binding is equally important for BrlR-DNA binding. The results show that all single-point mutations result in a mutant BrlR with lower DNA-binding capability compared with that of the wild-type BrlR (Fig. 4d, e and Supplementary Note 3). The results are consistent with the aforementioned c-di-GMP binding analyses (Supplementary Fig. 6a). The CD spectra of these mutants indicated that the impaired DNA-binding abilities were not due to structural mis-folding (Fig. 4f and Supplementary Fig. 6b).

We also performed β-galactosidase activity assays using a P*brlR-brlR-lacZ* reporter construct to investigate the transcriptional function of BrlR mutants in vivo. Functional analysis of these mutations showed that the c-di-GMP binding site mutants exhibited a clear decrease in the levels of BrlR-mediated activation of its own promoter. Under the current test conditions, BrlR mutants showed increased β-galactosidase activity at high c-di-GMP concentrations (Fig. 4g and Supplementary Note 3). To confirm the effect of c-di-GMP in *P. aeruginosa*, we carried out real-time quantitative PCR analyses. Following a previous report[5], we introduced the diguanylate cyclase PA4843 into a WT *P. aeruginosa* strain carrying either pHERD20T-empty or pHERD20T-PA4843. The transcription level of *brlR* was analyzed using the *brlR* 5 F and *brlR* 3 R primers (Supplementary Table 1).

The transcriptional level of *brlR* in pHERD20T-PA4843 strain is $4.5 \pm 1.3$-fold higher than that in the pHERD20T-empty strain (Fig. 4h).

**The BrlR-C domain adopts a new binding site for pyocyanin.** Structural comparisons demonstrate that the conserved BrlR-C domain has a drug-binding pocket similar to the one in SAV2435 (PDB code: 5KAU), indicating that BrlR-C domain may bind the small molecules structurally similar to RH6G that is able to bind SAV2435 (Fig. 5a). Further studies show that BrlR binds diverse toxic compounds in the same drug-binding pocket of the BrlR-C domain (Supplementary Note 5), e.g., fluorescein (the fluorophore of F-c-di-GMP), EB, and tobramycin (Fig. 5b–d and Supplementary Figs. 11,12). These findings explained why BrlR-C2E mut1 still binds F-c-di-AMP with a $K_D$ of $63.7 \pm 5.1\,\mu M$

(Fig. 2i), while raised a concern that the interaction between the fluorophore of F-c-di-GMP and the BrlR-C domain could significantly interfere with the measurement for the *c*-di-GMP binding sites. Indeed, fluorescein binds to BrlR with a $K_D$ of $33.6 \pm 1.6\,\mu M$ (Fig. 5c). Fortunately, F-c-di-GMP binds to the BrlR-C domain with a higher $K_D$ of $88.5 \pm 9.9\,\mu M$, probably due to its bigger size versus fluorescein (Supplementary Fig. 11). This $K_D$ could be roughly seen as a background in the FP assays. Because this value is much higher than that of BrlR binding to F-c-di-GMP ($K_D = 7.3 \pm 0.5\,\mu M$; Fig. 2g) or F-c-di-AMP ($K_D = 12.9 \pm 0.7\,\mu M$; Fig. 2i), we know that the $K_D$ value we determined is mainly contributed by the interaction between BrlR and *c*-di-GMP (or *c*-di-AMP), considering that only 1.0 nM F-c-di-GMP was used in FP binding assays.

Previous studies have shown that the addition of pyocyanin to *P. aeruginosa* PA14 strain elicits a 2.3-fold up-regulation of

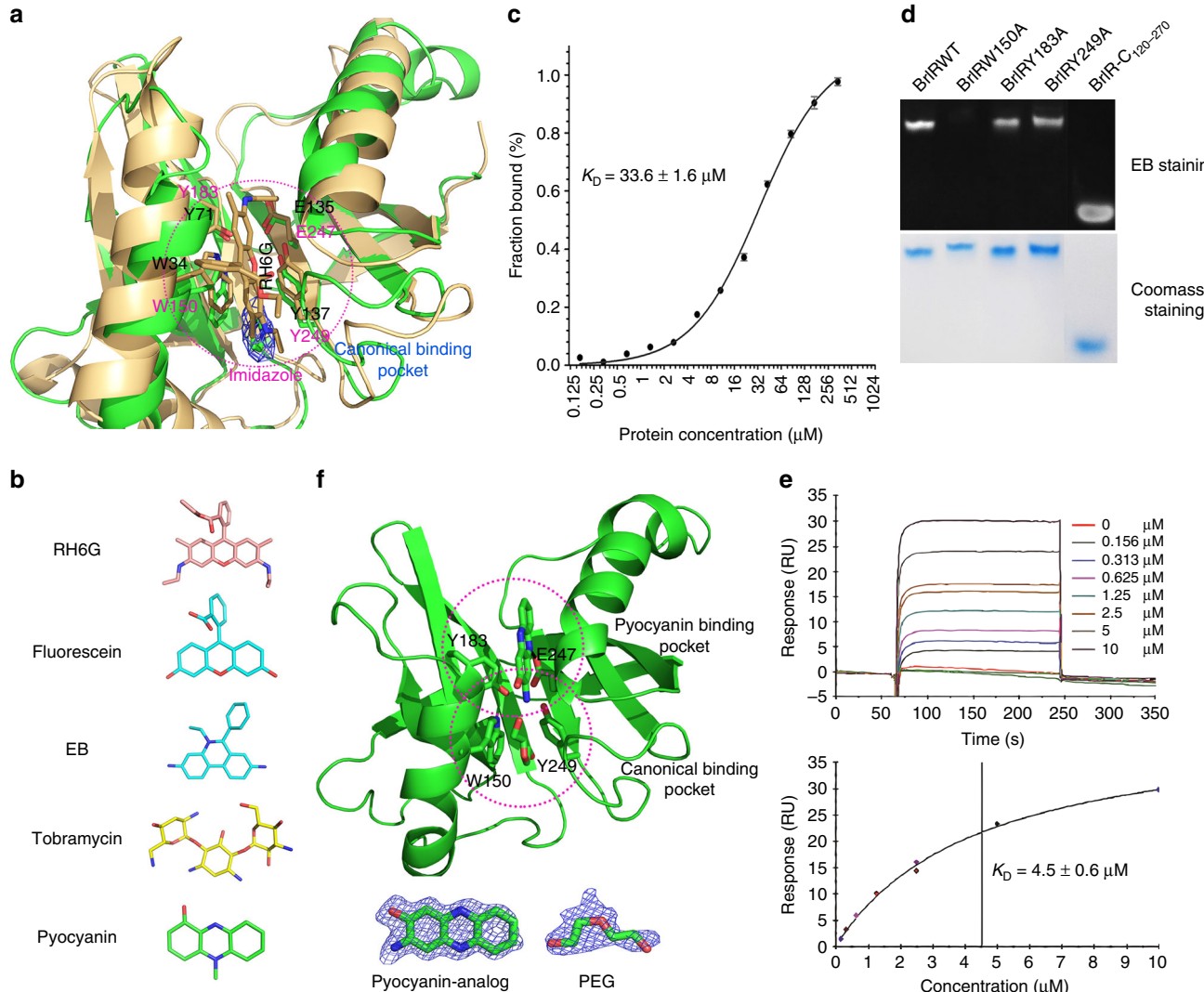

**Fig. 5** The BrlR-C domain can bind to diverse toxic compounds with two adjacent binding pockets. **a** Structural superpositions of BrlR-C and RH6G-bound SAV2435 (PDB code: 5KAU). The residues in contact with the ligand are shown as stick models and are labeled. Electron density (*2Fo - Fc* map) for imidazole (blue mesh) binding to the BrlR–c-di-GMP complex is contoured at the 1σ level. **b** The chemical structures of toxic compounds. **c** FP analyses of the fluorescein binding to BrlR. The binding curves were fit to deduce the binding affinities. **d** Two µg of BrlR variants were run on the native gel and stained with EB; the scan results are shown in the upper panel. The proteins were visualized by staining with Coomassie blue in the lower panel. **e** SPR sensorgram and the resulting affinity fit for pyocyanin binding to BrlR. The ligand binding and dissociation phases for all sensorgrams are shown in the upper panel. The concentrations of a pyocyanin are indicated. The binding responses were measured for 4 s before the end of the injection, and the $K_D$ values were calculated using the BIAevaluation software in the lower panel. **f** The monomeric structure of BrlR-C-3A2P complex. 3A2P, PEG and the conserved residues are shown in stick mode. Below the structures are *Fo–Fc* omitted electron density maps contoured at 3.0σ for pyocyanin and PEG

_brlR_[35]. Considering that pyocyanin is structurally similar to the backbone of the fluorophore and EB[51] (Fig. 5b), we hypothesized that the BrlR-C domain might bind pyocyanin. The surface plasmon resonance analysis (SPR) results confirmed our hypothesis and showed that BrlR bound pyocyanin tightly with a $K_D$ of 4.5 μM (Fig. 5e), which was ~140-fold higher than the binding affinity of tobramycin ($K_D$ = 0.64 mM, Supplementary Fig. 12a). To clarify how BrlR specifically recognizes pyocyanin, we

crystallized the BrlR or BrlR-C domain in the presence of both pyocyanin and 3A2P. Despite great effort, we were unable to obtain diffraction-quality crystals of the BrlR-pyocyanin complex. Finally, the structure of the BrlR-C domain in complex with 3A2P was successfully determined (Fig. 5f). Surprisingly, the electron density shows that 3A2P does not bind to the same position as other ligands do in MerR family proteins (Fig. 5a–f). 3A2P is located proximal to the two helices and next to the canonical

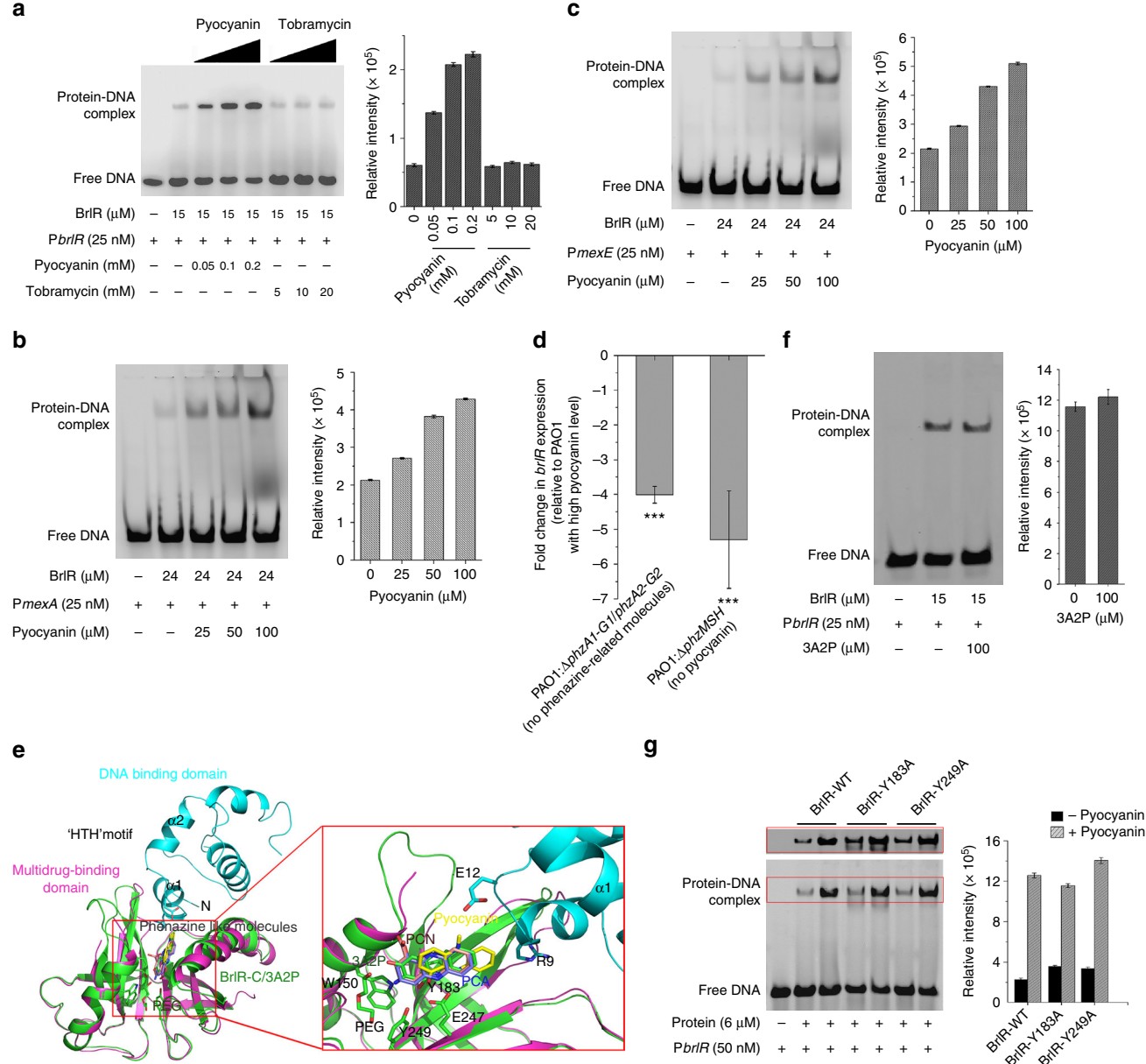

**Fig. 6** BrlR is a pyocyanin responsive DNA binding protein. **a** BrlR-DNA binding assays using FAM-labeled P*brlR* DNA (25 nM) in the increasing concentrations of pyocyanin and tobramycin. A total of 225 pmol of BrlR was used. **b**, **c** BrlR binding to P*mexA* (**b**) and P*mexE* (**c**) was enhanced in the presence of increasing concentrations of pyocyanin. The protein-DNA complexes were quantified by band densitometry (right). Error bars, s.d., obtained from triplicate experiments. **d** qRT-PCR demonstrating decreased brlR transcript levels in Δ*phzA1*-G1/*phzA2*-G2 strain (not producing phenazine) and Δ*phzMSH* strain (not producing pyocyanin). The fold changes in the *brlR* transcript level were calculated relative to the *brlR* transcript in strain PAO1 at high pyocyanin concentrations. All values represent means ± s.d. obtained from five independent experiments performed in duplicate. ***significantly different from PAO1 ($P \leq 0.001$, one-tailed *t*-test). **e** The docking results of pyocyanin and its analogs at the unique interface of apo-BrlR. Close view of the pyocyanin binding pocket for the top 10 results and the solved pyocyanin analog (3A2P) shown in the red rectangle. The related residues referred to in the text are labeled. **f** EMSA for BrlR binding to P*brlR* in the absence or presence of 3A2P. **g** EMSA for the mutants of pyocyanin-binding residues that bind to P*brlR* in the absence or presence of pyocyanin. Protein concentrations are the same. The results were quantified by band densitometry (right). Error bars, s.d., obtained from triplicate experiments

ligand-binding pocket of the BrlR-C domain. The newly identified binding site is unique among those of all GyrI-like structures determined to date[52]. The ligand is stabilized chiefly by hydrophobic interactions with the surrounding residues in addition to an H-bond with E247 at the bottom. A fragment of diethylene glycol (PEG) is also observed in the structure of the 3A2P-BrlR-C domain within the pocket corresponding to the RH6G-binding site in SAV2435 (Fig. 5f). This result indicates that the BrlR-C domain can bind to pyocyanin and other ligands simultaneously, which is consistent with the multiple binding modes identified in the TetR family QacR[53,54]. The residues lining the pyocyanin binding pocket are conserved in many GyrI-like proteins (Supplementary Fig. 5b), suggesting that the GyrI-like domains in other bacteria may also recognize pyocyanin. Together, the BrlR-C domain contains two adjacent binding sites within the multidrug-binding pocket[55].

**BrlR is a pyocyanin-responsive transcriptional regulator.** The BrlR-C domain has been demonstrated to bind pyocyanin efficiently. We next sought to verify whether pyocyanin could act as a transcriptional activator to enhance BrlR-DNA binding. EMSAs were conducted in the absence or presence of pyocyanin and tobramycin. Whereas little BrlR-DNA binding was detected in the absence of these compounds, addition of pyocyanin resulted in a significant increase in BrlR-DNA binding without affecting BrlR-DNA migration in the gel (Fig. 6a). Increasing pyocyanin concentrations lead to more BrlR-DNA complex formation, indicating that pyocyanin enhances the BrlR-DNA binding in a concentration-dependent manner, just like c-di-GMP. Enhanced DNA binding was not limited to PbrlR, because BrlR binding to its target promoters PmexA and PmexE (Sequence in Supplementary Table 1), was also enhanced by the presence of pyocyanin (Fig. 6b, c). In contrast, the addition of tobramycin did not result in an observable difference in the BrlR-PbrlR binding at the current concentrations. Thus, tobramycin is not a specific

activator of BrlR like pyocyanin (Fig. 6a). To further understand the functions of pyocyanin and c-di-GMP in regulating brlR gene expression in vivo, real-time quantitative PCR analysis was performed for *P. aeruginosa* PAO1, *P. aeruginosa* PAO1:ΔphzMSH, and *P. aeruginosa* PAO1:ΔphzA1-G1/phzA2-G2 strains. The results show that the expression level of the brlR gene in the PAO1 strain was ~4.0-fold higher than that in a ΔphzA1-G1/phzA2-G2 strain not producing phenazine and ~5.3-fold higher than that in a ΔphzMSH strain with no pyocyanin production (Fig. 6d). While the ΔphzMSH strain still produces phenazine-1-carboxylic acid (PCA)[56], which is a pyocyanin analog, the expression level of the brlR gene was even lower in the presence of PCA production (Fig. 6d). Moreover, in the susceptibility assay, addition of pyocyanin renders planktonic wild-type PAO1 cells but not ΔbrlR cells resistant to tobramycin (Fig. 7a). After treatment with tobramycin, the CFU number of PAO1 decreases by ~3.4-order of magnitude (log reduction) in the absence of pyocyanin, while the number only decreases by ~1.5-order of magnitude in the presence of pyocyanin. In the background of ΔbrlR, no significant difference was observed between the viabilities of tobramycin-treated cells with and without addition of pyocyanin. Interestingly, for the cells harboring pHERD20T-PA2133 that contain low c-di-GMP levels[5], an elevated level of pyocyanin is still able to increase their tobramycin resistance (Fig. 7a). The addition of pyocyanin decreases the susceptibility of plasmid-harboring cells from a 4.3-order of magnitude reduction to a 1.7-order of magnitude reduction in CFU number caused by tobramycin treatment.

Based on the structure of the 3A2P-BrlR-C domain, we modeled pyocyanin and other phenazine-like molecules in apo-BrlR by using the AUTODOCK program (version 4.2)[57]. The calculated complex structure reveals the interactions between BrlR and pyocyanin in detail (Fig. 6e). The docked molecules are expected to be situated in the hydrophobic pocket in which 3A2P also binds. The position of pyocyanin is close to the interface of two terminal domains of BrlR. The side chain of E247 stabilizes

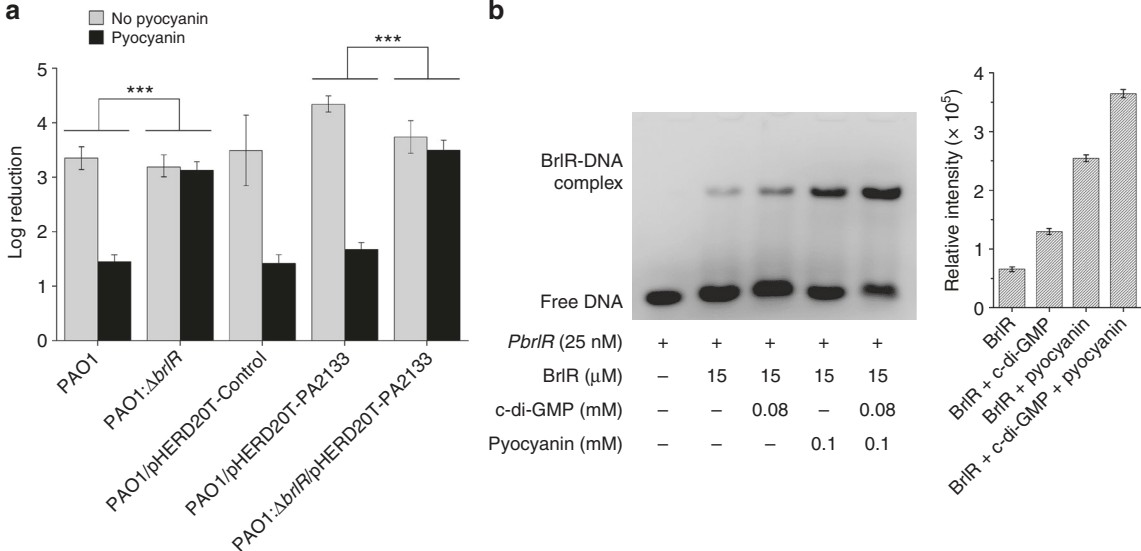

**Fig. 7** The correlation between pyocyanin and c-di-GMP in regulation of BrlR. **a** Elevated pyocyanin and decreased c-di-GMP levels render the planktonic *P. aeruginosa* PAO1 cells, but not ΔbrlR cells, resistant to tobramycin. All strains were grown planktonically to the mid-log phase in the presence or absence of 0.2 mM pyocyanin, which was purified from stationary phase cultures of PAO1. Susceptibility of *P. aeruginosa* strains was determined by the treatment of 50 μg/ml tobramycin for 1 h. Strain PAO1 harboring the empty plasmid pHERD20T was established as a control, and strain PAO1 or ΔbrlR containing pHERD20T-PA2133 was shown to contain a low level of c-di-GMP[5]. Error bars, s.d., obtained from triplicate experiments. The data were analyzed with a one-tailed *t*-test (***$P \leq 0.001$). **b** c-di-GMP and pyocyanin can cooperatively enhance BrlR-DNA binding. EMSA for the interaction between FAM-labeled PbrlR DNA and BrlR in the presence of c-di-GMP, pyocyanin, and both. The concentration of each component is indicated. The BrlR-DNA complexes were quantified by band densitometry (right). Error bars, s.d., obtained from triplicate experiments

pyocyanin through an H-bond. Compared with 3A2P, the phenazine ring of pyocyanin slides aside slightly and pushes against the R9 and E12 side chains of the HTH motif. Thus, we speculate that the HTH motif undergoes significant movement to facilitate BrlR-DNA binding when pyocyanin binds to BrlR. EMSA results showed that, compared with pyocyanin, 3A2P cannot enhance BrlR-P*brlR* binding at the same concentration (Fig. 6f). Other phenazine-like molecules, such as PCA and phenazine-1-carboxamide (PCN), are all expected to be located in exactly the same position as 3A2P (Fig. 6e), therefore, the molecules might not induce a conformational change in the HTH motif and exhibit a minor effect on BrlR-DNA binding compared with that of pyocyanin. Indeed, PCA could not up-regulate *brlR* transcription in strain PAO1 (Fig. 6d). These results indicate that pyocyanin is a well-chosen inducer for BrlR.

Structural analysis showed that the position and conformation of the conserved residue side chains lining the two adjacent binding pockets remain largely the same before and after 3A2P binding. Hence, the pyocyanin binding pocket of BrlR-C domain is quite rigid. It is unlikely that the larger drug molecules located in the canonical binding pocket can slide into the pyocyanin binding site and promote BrlR-DNA binding, as reflected by the EMSA results for tobramycin (Fig. 6a). To further verify the amino acid residues contributing to pyocyanin binding, many BrlR mutants were constructed. Whereas mutant E247A was insoluble, mutant F253R was not expressed, and mutant C173W was unstable. The binding affinity of Y183A and Y249A for pyocyanin were evaluated by using EMSA with a 36-bp P*brlR*. Both Y183A and Y249A exhibited a slightly weaker response to pyocyanin binding (Fig. 6g).

## Discussion

BrlR is a member of MerR-family transcriptional regulators from *P. aeruginosa*, which is activated by c-di-GMP[5]. Recently, an increasing number of c-di-GMP-responsive transcriptional regulators have been identified[26], including VpsT, LtmA, FleQ, MrkH, and BldD[20–24]. Our results provide a good example of an HTH motif containing two separate binding sites, and each site respectively binds one c-di-GMP molecule in a stacked conformation (Fig. 2a). Intriguingly, although *P. aeruginosa* does not produce c-di-AMP, the first c-di-GMP binding site of BrlR also recognizes c-di-AMP (Fig. 2i), which not only helps to understand c-di-AMP binding to the proteins closely related to BrlR in the c-di-AMP-producing bacteria but also raises the possibility that BrlR homologs could function as cyclic-di-nucleotide receptors in both Gram-negative and Gram-positive bacteria.

Our work also provides a structural basis for the specific recognition of c-di-GMPs by BrlR. However, many questions remain to be answered. Our efforts to crystallize the BrlR-DNA complex and the ternary complexes with different ligands failed. Hence, we do not know how BrlR affects the conformation of its target promoter DNA to regulate transcription. MerR-family members share a high degree of structural homology in their HTH motif and normally act as a dimer to regulate transcription by DNA untwisting[49]. We suggested that BrlR is a stable tetramer in solution (Fig. 1g), while the crosslinking assays have shown that BrlR is predominantly a dimer[5]. The inconsistence of the observations from different assays might be due to the nature of cross-linking reaction and the dimer-of-dimer organization of BrlR (Supplementary Note 6, Supplementary Fig. 13). In a BrlR dimer, the distance between two recognition helices (α2) of the HTH motif is longer than that in the classical BmrR dimer (Supplementary Fig. 14a, b). The blocked HTH motif and the long distance between two recognition helices hinder BrlR to approach and bind DNA, thus resulting in a very low DNA-

binding capability of BrlR (Fig. 4a). The recognition helix (α2) of the HTH motif is 1.2 Å shorter ($C^\alpha$(H20)–$C^\alpha$(H20′)) in the BrlR-c-di-GMP complex than that in apo-BrlR (Supplementary Fig. 14a), which could help BrlR approach DNA. In addition, two c-di-GMP molecules binding to the HTH motif are close to the putative DNA sequence and thus may facilitate DNA binding (Supplementary Fig. 14c). As might be expected, the bound DNA fragment would interact with the blocked surface of the HTH motif of BrlR. Thus, the adjacent BrlR-C domains must move away (Supplementary Fig. 14c, d). It is possible that the promoter DNA binding leads to separation of the BrlR tetramer into two dimers. During this process, the BrlR dimer undergoes tremendous conformational changes to twist the length of DNA, which allows for efficient binding by RNA polymerase. However, without the crystal structure of BrlR in complex with DNA, the detailed mechanism remains unknown.

Pyocyanin is an important virulence factor secreted by *P. aeruginosa*. Recent work has shown that pyocyanin plays important roles in both *P. aeruginosa* biofilm formation[33] and multidrug efflux pumps expression[34,35]. The complex regulative function of pyocyanin requires the existence of multiple intracellular receptors to mediate its function. However, despite pyocyanin's long history, the molecular mechanism of its receptors is unclear. Here, BrlR is identified to be among the pyocyanin receptors reported to date. Based on the structure of 3A2P-BrlR-C, more pyocyanin-like molecules are docked. Among these ligands, pyocyanin is the only one expected to affect the interface of the two terminal domains in BrlR (Fig. 6e), and only the binding of pyocyanin to the corresponding site will induce a significant conformational change in the HTH motif. Therefore, pyocyanin is a specific inducer for BrlR compared with other pyocyanin analogs, which is supported by our qRT-PCR results (Fig. 6d). Various antibiotics, such as tobramycin and gentamicin, bind to the BrlR-C domain like to BmrR[11,44] (Supplementary Note 5). However, their binding affinities are very low (Supplementary Fig. 12). In addition, tobramycin does not enhance the BrlR-P*brlR* binding as pyocyanin does (Fig. 6a). Therefore, the binding of antibiotics to BrlR might have no physiologic significance.

Our most striking finding is that BrlR is not only a pyocyanin receptor but it is also a dual receptor that is responsive to both c-di-GMP and pyocyanin. Binding of both c-di-GMP and pyocyanin enhances BrlR-DNA binding and gene transcription. c-di-GMP molecules mainly interact with the 'wing' loop and helix α4 in the DNA-binding domain, whereas pyocyanin has extensive interactions with the HTH motif of BrlR once it binds to the BrlR-C domain. Intriguingly, simultaneous binding of the low concentration of c-di-GMP and pyocyanin renders BrlR-DNA binding more efficient, as demonstrated by EMSA results (Fig. 7b). Hence, it is interesting to us whether the level of pyocyanin or c-di-GMP contributes to antimicrobial-agent resistance in *P. aeruginosa* cells. Our results showed that a high level of pyocyanin renders planktonic wild-type and pHERD20T-PA2133-harboring cells but not Δ*brlR* cells resistant to tobramycin treatment. It has been shown that a high level of c-di-GMP increases cell resistance to antibiotics in *P. aeruginosa*[43], and overexpression of PA2133 by plasmid pHERD20T-PA2133 decreases c-di-GMP level[5]. In cells with pHERD20T-PA2133, the addition of pyocyanin is still able to increase tobramycin resistance despite the cells' low level of c-di-GMP. These results suggest that an elevated level of pyocyanin renders *P. aeruginosa* resistant to tobramycin via a *brlR*-dependent mechanism.

Biofilm formation is often accompanied by the expression of drug resistance genes, thus representing a possible evolutionary advantage for bacteria. In the natural aquatic habitat, there are some microbes that produce antibiotics, but the concentration of the antibiotic cannot achieve an effective concentration

before being diluted into the water. Over time, the antibiotic-producing microbes adapt to avoid wasting energy by making antibiotics in aquatic environment. The potential victims also adapt to avoid wasting energy on the expression of anti-drug genes. For cells in biofilms, however, the situation is quite different. Extracellular matrix of bacterial biofilms has been shown as a diffusion barrier for antibiotics[58], thus antibiotics may accumulate to high concentrations[59]. The potential victims must defend against the adverse environment, thus potentially explaining why biofilm formation and drug-resistant genes are tightly and cooperatively regulated. It is not surprising that both c-di-GMP and pyocyanin regulate multiple physiological processes. Strikingly, BrlR is a receptor that is responsive to both c-di-GMP and pyocyanin, thus indicating that these two signaling pathways crosstalk and that BrlR establishes a connection between c-di-GMP, pyocyanin and the expression of multidrug efflux pumps. Moreover, BrlR acts as a pyocyanin receptor. Because pyocyanin regulates many cellular functions, there must be many unknown receptors remaining to be identified. The identification of BrlR as a pyocyanin receptor should provide useful knowledge to guide future research.

## Methods

**Bacterial strains, plasmids and culture conditions**. The strains and plasmids used in this study are listed in Supplementary Table 2. Unless otherwise indicated, all strains were grown in Luria Bertani broth (LB, Becton Dickinson) medium. In particular, to detect the influence of pyocyanin, all PAO1 strains were grown on Pseudomonas minimal medium (PMM) (Contributed by Robert L. Wick) at 37 °C with shaking at 150 rpm. When necessary, strains were cultured in medium supplemented with appropriate antibiotics at the following concentrations: ampicillin (Amp; 100 µg/ml), kanamycin (Kan; 50 µg/ml), carbenicillin (Car; 300 µg/ml for *P. aeruginosa*), and tetracycline (Tet; 25 µg/ml for *E. coli* and 120 µg/ml for *P. aeruginosa*).

**Expression and purification of BrlR protein**. The plasmid encoding the His$_6$-Sumo-BrlR was constructed and introduced into *E. coli* strain BL21 DE3. The recombinant strains were incubated in LB medium supplemented with 50 µg/ml kanamycin at 37 °C to an A$_{600}$ of 0.8. After 4 h of induction with 0.12 mM iso-propyl-D-1-thiogalacto-pyranoside (IPTG) at 37 °C, the cells were harvested at 6000×g for 15 min at 4 °C. The pellet was resuspended in 25 mM Tris-HCl buffer at pH 8.0 containing 200 mM NaCl and was lysed by sonication. The lysate was then centrifuged at 28,370×g for 50 min at 4 °C, and the supernatant was loaded onto a Ni-NTA column (GE Healthcare) for affinity chromatography. His$_6$-SUMO-tagged protein was eluted with 25 mM Tris-HCl buffer at pH 8.0 containing 100 mM NaCl and 250 mM imidazole. The eluted protein was incubated with the protease UlpI at a 100:1 ratio (w/w) and dialyzed against the reaction buffer (10 mM Tris-HCl, pH 8.0, 100 mM NaCl) at 4 °C overnight. The cleavage mixture was loaded onto Ni-NTA column again to remove the cleaved His$_6$-SUMO tag and the His$_6$-UlpI tag. Subsequently, untagged BrlR protein was purified by ion exchange chromatography (Source Q; GE Healthcare) and size exclusion chromatography (Superdex 200; GE Healthcare) with 10 mM Tris-HCl buffer at pH 8.0 containing 100 mM NaCl. All the BrlR mutant proteins were then purified with the same protocol used for native protein. For BrlR-C (120-end) protein purification, the purification protocol was almost the same, except the induction temperature was decreased to 18 °C, and Ppase was used to remove His$_6$-tag.

**Crystallization, data collection, and structure determination**. The protein used for Crystallization screens was 10 mg/ml and incubated with c-di-GMP at 1:5 molar ratio for co-crystallization experiments. BrlR-C domain of 14 mg/ml was incubated with a pyocyanin analog (3-amino-2-phenazino, 3A2P) at 1:3 ratio for Crystallization screens. Hampton Research kits were used in the hanging drop vapor diffusion method to get preliminary crystallization conditions. BrlR crystal was obtained in 1.8 M sodium acetate (pH 7.0), 0.1 M Bis-Tris propane (pH 7.0) after one week at 20 °C. The crystal of BrlR-c-di-GMP complex crystalized in a solution containing 0.1 M Lithium sulfate, 0.1 M Sodium citrate tribasic (pH 5.5) and 20% (w/v) PEG,1000 after 3 days at 20 °C. The crystal of BrlR-C (120-end) bound to pyocyanin analog was obtained in 0.1 M Bis-Tris (pH 5.5) and 25% (w/v) PEG,3350 after 3 days at 20 °C. To solve phase problem, the heavy-atom reagent I3C (Hampton Research) was soaked into the BrlR/c-di-GMP complex crystals by addition of 0.5 µl 0.5 M I3C solution directly to the crystallization drop. After 30 min, crystals were picked out to use. All crystals were flash frozen in liquid nitrogen, with addition of 20% (v/v) glycerol as cryoprotectant.

X-ray diffraction data were collected at the beamline BL17U1 of the Shanghai Synchrotron Radiation Facility (SSRF)[60]. All data were processed using HKL-2000[61]. The original structure of BrlR-c-di-GMP was determined by using the single-wavelength anomalous dispersion (SAD) phasing. Phases were calculated using AutoSol implemented in PHENIX[62]. AutoBuild in PHENIX was used to automatically trace the chain of BrlR-c-di-GMP. MR was then performed with this model as a template by using MOLREP in CCP4 to determinate the structure at higher resolution[63]. The structure aop-BrlR and BrlR-C bound to 3A2P were determined by using MR with Phaser, with the complete and partial c-di-GMP-bound BrlR structure as the search models respectively. After several rounds of positional and B-factor refinement using Phenix.Refine with TLS parameters[62] alternated with manual model revision using Coot[64], the quality of final models were checked using the PROCHECK program[65]. Details of the data-collection and refinement statistics are given in Table 1. All of the figures showing structures were prepared with PyMOL (http://www.pymol.org).

**Quantitative reverse transcriptase PCR (qRT-PCR)**. PAO1 WT strains were grown under different conditions. For c-di-GMP effects, overexpression of PA4843 in PAO1 (PAO1/pHERD20T-PA4843) relative to WT cells (PAO1/ pHERD20T) was determined. When *P. aeruginosa* strains entered mid-log phase, arabinose was added to induce gene expression. To test pyocyanin's effect, a *P. aeruginosa* PAO1 WT strain cells with high pyocyanin production was determined relative to a Δ*phzA1-G1/phzA2-G2* strain not producing phenazine and a Δ*phzMSH* strain not producing pyocyanin.

Total RNA was extracted using a MiniBEST Universal RNA Extraction Kit (TaKaRa) and cDNA synthesis was performed using a PrimeScript RT reagent Kit (TaKaRa). Primers for RT-PCR listed in Supplementary Table 1 were designed using the program PrimerExpress (Applied Biosystems). The RT-PCR reactions were performed on a CFX96 Real-Time PCR Detection System (Bio-Rad) using SYBR Premix Ex Taq™ (TaKaRa). Expression of the target genes was normalized to the expression of 16 S RNA. The relative transcript abundance was calculated according to the $2^{-\Delta\Delta Ct}$ method[66]. Five biological replicates were performed per sample. The data were analyzed with a one-tailed *t*-test (***$P \leq 0.001$).

**Surface plasmon resonance analysis**. The binding affinity of diverse toxic compounds for BrlR were investigated by SPR using a BIAcore-T200 apparatus (GE Healthcare). The ligand, BrlR (20 µM in 10 mM sodium acetate, pH 4.5), was immobilized on an activated CM5 sensor chip by amine coupling (BIAcore) with 4400 response units (RU), and the unreacted sites were blocked with 1.0 M ethanolamine–HCl (pH 8.5). Steady-state binding analysis was performed with injections of different molecules at different concentrations over the immobilized BrlR surface at a flow rate of 20 µL min$^{-1}$ at 25 °C. Association was measured for 180 s, followed by 120 s of dissociation in running buffer (20 mM Hepes pH 7.5, 150 mM NaCl). The sensorgrams for specific interactions were obtained by subtracting the reference unimmobilized flow cell response from that of the sample. Since BrlR only has one binding site for pyocyanin or other toxic compounds (Fig. 5a–f), the kinetic analysis of the sensorgrams from analyte interaction with the immobilized BrlR was performed by fitting the binding curves to a 1:1 binding model. To determine the steady state $K_D$ values, equilibrium responses were plotted against analyte concentrations, and the resulting curves were fitted to the steady state model in the BIAevaluation software package, version 3.0 (GE Healthcare).

**Fluorescence polarization measurements**. F-c-di-GMP and F-c-di-AMP both contain a fluorophore at one of the 2′-ribose hydroxyls of c-di-GMP and c-di-AMP, respectively (Fig. 2f). To measure the binding of c-di-GMP and c-di-AMP to BrlR variants, 1 nM F-c-di-GMP or F-c-di-AMP was incubated with increasing amounts of indicated BrlR WT, BrlR mutation, or BrlR-C in reaction buffer (25 mM HEPES, 100 mM NaCl, pH 7.5) at 25 °C for 20 min. 1 nM fluorescein was incubated with BrlR WT in the same reaction buffer at 25 °C. FP measurements were conducted on a Synergy 4 Microplate Reader (BioTek) at 25 °C. All experiments were performed in triplicate. The curves were fitted to deduce binding affinities by GraphPad Prism 5.

To determine the stoichiometry of c-di-GMP binding to BrlR, 1 nM F-c-di-GMP was mixed with 160 µM c-di-GMP (which is 20-fold higher than the $K_D$ of BrlR-bound F-c-di-GMP) and incubated with increasing amounts of BrlR. The resulting data exhibited a linear increase in the observed millipolarization until the saturation of c-di-GMP by BrlR, and the millipolarization values remained unchanged after saturation. The binding stoichiometry of c-di-GMP to BrlR was calculated as the initial concentration of c-di-GMP (160 µM) divided by the BrlR concentration (88 µM) at the inflection point. To determine the stoichiometry of c-di-AMP binding to BrlR, 1 nM F-c-di-AMP mixed with 195 µM c-di-AMP (15-fold higher than the $K_D$ of BrlR bound F-c-di-AMP) was used in the same assay, and the stoichiometry was calculated as 190 µM c-di-AMP to 195 µM BrlR.

**Electrophoretic mobility shift assay**. A 6-carboxy-fluorescein (FAM)-labeled 36-bp *brlR* promoter DNA (P*brlR*) or 80-bp *mexA/mexE* promoter DNA (P*mexA*/P*mexE*) probes (~0.375 pmol), BrlR protein and varying amounts of inducers were incubated in binding buffer (10 mM Tris HCl [pH 7.4], 50 mM KCl, 1 mM DTT, 100 mg/mL BSA and 5 ng/mL poly [dI-dC]) for 60 min at 25 °C. Finally, glycerol was added to a concentration of 6.5%. The samples were subjected to a 5% poly-acrylamide gel electrophoresis and run in 0.5 × Tris-borate-EDTA buffer at 80 V

for 1 h at room temperature. Imaging and data analyses were performed using a Typhoon Scanner (GE Healthcare) and Imagequant software. The probes used are listed in Supplementary Table 1.

**β-galactosidase assay**. β-galactosidase activity was assayed with *P. aeruginosa* PA01 Δ*brlR* strain carrying plasmid pME6522-borne transcriptional lacZ fusions[67]. *P. aeruginosa* carrying the empty vector was used as the positive control. The reporter plasmids are derivatives of the pME6522 plasmid in which multiple cloning sites were inserted before the LacZ with the promoter and coding regions of *brlR* with diverse mutations. These reporter plasmids were transformed into the reporter strain. Overnight cultures were diluted 20-fold with LB medium supplemented with appropriate antibiotics. The bacterial cultures were grown at 37 °C in with shaking until the their OD600 reached 0.6–0.8, after which the production of β-galactosidase was assayed in triplicate for each sample through the Miller method[68].

**Planktonic antibiotic susceptibility testing**. To determine the role of pyocyanin in antimicrobial susceptibility, *P. aeruginosa* strains (Supplementary Table 2) were grown at 37 °C in 20-fold diluted LB medium supplemented with or without 0.2 mM pyocyanin, which was purified from stationary phase cultures of PAO1. Tobramycin was added to the mid-log phase ($OD_{600} = 0.8$) cells suspension to a final concentration of 50 μg/ml; after 1 h, the broth was serially diluted and spread-printed onto LB agar (medium containing 300 μg/ml carbenicillin was used for the cells harboring plasmid) to determine the cell viability via CFU counts. The cell suspensions without tobramycin were set as controls. Susceptibility is expressed as a logarithmic reduction in CFU amount. The data were analyzed with a one-tailed *t*-test (***$P \leq 0.001$).

**Molecular docking of pyocyanin and its analogs**. The program AutoDock 4.2[69] was used for the molecular docking calculations. The structure of BrlR-C bound to 3A2P was used as the reference. Pyocyanin and other pyocyanin analogs were prepared using AutoDock Tools. All rotatable bonds of the ligands were accepted, and the receptor was assumed to be rigid. The docking calculations were carried out following a detailed protocol[57]. A composite file of ten possible conformers was analyzed and their positions and conformations are almost the same.

**Gel-filtration assay**. The BrlR and BrlR with c-di-GMP or pyocyanin were subjected to gel-filtration analysis (Superdex 200 10/300 GL column; GE Health-care; 10 mM Tris–HCl pH 8.0; 100 mM NaCl) in the presence and absence of ligand, the Superdex buffer contains 50 μM c-di-GMP or 5 μM pyocyanin for the BrlR/ligand complex, respectively. The assay was performed at a flow rate of 0.4 ml min$^{-1}$ and 0.1 ml of BrlR (about 1.0 mg ml$^{-1}$) was injected at 4 °C. The calibration curve was generated using the standard proteins to indicate the molecular mass (in kDa).

**Analytical ultracentrifugation**. Sedimentation velocity experiments were performed in a ProteomeLab XL-I analytical ultracentrifuge (Beckman Coulter, Brea, CA), equipped with AN-60Ti rotor (4-holes) and conventional double-sector aluminum centerpieces with a 12 mm optical path length, loaded with 380 μL of samples and 400 μL of buffer (20 mM Tris·HCl, pH 8.0, 100 mM NaCl). The rotor was first equilibrated for ~1 h at 20 °C in the centrifuge. In addition, experiments were carried out at 20 °C and 36,000 rpm, using a continuous scan mode and a radial spacing of 0.003 cm. Scans were collected at 3-min intervals at 280 nm. The fitting of absorbance versus cell radius data was performed using SEDFIT software (https://sedfitsedphat.nibib.nih.gov/software) and a continuous sedimentation coefficient distribution c(s) model, covering a range of 0–15 S. The buffer composition (density and viscosity) and protein partial specific volume (V-bar) were calculated using SEDNTERP: buffer density $\rho = 1.0193$ g/cm3, viscosity $\eta = 0.01041$.

**Chemical cross-linking**. BrlR protein (20 μM) in cross-linking buffer (25 mM Hepes pH 8.0 and 150 mM NaCl) was incubated with a freshly prepared solution of 1 mM dithiobis (succinimidyl propionate)-DSP at room temperature for 1, 5, 10, or 30 min. The reactions were quenched by the addition of 50 mM Tris·HCl (pH 8.0), and the cross-linked products were analyzed by SDS-PAGE.

**Data availability**. Atomic coordinates and structure factors have been deposited in the PDB with accession codes 5XBW, 5XBT and 5XBI. All other relevant data are available in this article and its Supplementary Information Files, or from the corresponding authors upon request.

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

## Acknowledgements

We thank Xu Feng and Haixin Yuan (FuDan University, School of Medicine) for EMSA experiments, Chao Gao (Shandong University, School of Life Science) for providing pVS1-p15A shuttle vector, Jing He, Maofeng Wang and Xiaotong Diao (Shandong University, School of Life Science) for help in protein purification and biochemical assays. This study was supported by the National Basic Research Program of China (973 Program) 2015CB150600 to L.G., the National Natural Science Foundation of Chi`na Grant 31470732 (to L.G.) and 31370110 (to W.H.), and the Shandong International S&T Cooperation Demonstration Program 2017JHZ009 (to W.H.).

## Author contributions

F.W., Q.H., and L.G. designed, performed, and analyzed experiments, and wrote the manuscript; F.W. and Q.H. collected the data and determined the crystal structures. W.H. analyzed the data and revised manuscript. J.Y. performed the *brlR* gene knockout. S.X. analyzed the data. L.G. supervised the project.

## Additional information

**Competing interests:** The authors declare no competing interests.

