## [Peer Review File · Nature Communications]

Reviewers' comments:

Reviewer #1 (Remarks to the Author):

The manuscript "BrIR from *Pseudomonas aeruginosa* is a common receptor for both cyclic di-GMP and pyocyanin" by Wang and colleagues examines potential binding partners for the transcriptional regulator BrIR. The authors use a structural approach to examine binding of the BrIR protein to c-di-GMP, including identifying specific amino acids that directly interact with c-di-GMP. After identifying two c-di-GMP binding sites, the authors then examine how these sites impact binding of BrIR to target promoter regions and find that both sites are required for optimal DNA binding. Unusually for a MerR-family protein, BrIR appears to form a tetramer in the absence/presence of effector molecules, with minimal structural changes noted upon binding c-di-GMP. The conformational changes do not appear to be sufficient to free up predicted DNA binding domains. In addition, data was presented indicating pyocyanin also binds to BrIR and affects the efficacy of BrIR:DNA interactions. The authors present a compelling, well-organized case for the importance of both c-di-GMP and pyocyanin in modulating the ability of BrIR to bind to DNA target regions. However, the study also raised several concerns as detailed below.

Comments

1. Compared to other c-di-GMP binding proteins and the K_d for the *P. aeruginosa* BrIR reported in the literature, the K_d reported here is 1000-fold higher. The high K_d value was obtained using non-physiological conditions, as it would be unlikely that as much as 3 mM c-di-GMP would be present in a bacterial cell. Similarly, the K_d value for pyocyanin is also higher than expected (and may imply binding under non-physiological conditions as well). It would be useful to see if efficient binding could be obtained and studying using lower concentrations of potential effector molecules, or better expression why such high concentrations were utilized (see next comment).
2. Is the high K_d value related to the method or the BrIR protein itself? It may be beneficial to compare the K_d of known c-di-GMP protein (other than BrIR) using the method described in the manuscript. If the method is solid, this raises some serious concerns about the BrIR protein and whether the variants produced in *E. coli* are the real native structure.
3. The authors demonstrate that the apo-form of BrIR binds to DNA. This is a surprising finding that should warrant further testing as it may indicate lack of specificity. Therefore, I strongly suggest testing BrIR DNA binding using control promoters (e.g. those believed not to be linked to BrIR).
4. The specificity of binding to BrIR could be clarified by testing additional compounds related to c-di-GMP and pyocyanin. For example, cAMP could be tested to see if BrIR binds to cyclic nucleotides in general or c-di-GMP specifically. Likewise, other non-pyocyanin phenazines could determine BrIR specificity for pyocyanin.
5. The hypothetical models for the c-di-GMP-responsive binding to BrIR seem to be rather speculative. It would be useful to perform additional experiments to elucidate the response to c-di-GMP binding or eliminate them entirely.
6. The authors indicate that BrIR, unlike other MerR proteins, forms tetramers (dimers of dimers). Given the unusual oligomerization status of BrIR and to ensure that this is not an artifact due to protein overproduction in *E. coli*, the authors should consider determining whether the tetramer is detectable in *P. aeruginosa*.

7. It would be interesting to see if BrIR could be crystalized in the presence of both c-di-GMP and the pyocyanin analogue 3A2P. This would provide valuable information about potential steric hindrance caused by binding to one ligand versus another.

8. The mutants Y183A and Y249A are said to exhibit a weak response to pyocyanin (L. 311-313), yet Fig.6 seems to suggest that the response is the same as for the wild-type protein.

9. It would be beneficial to expand the qPCR experiment regarding DGC PA4843 overexpression to include the mutant BrIR variants, or to repeat it with the Miller assay strains.

10. While conditions correlating with higher pyocyanin production were presented, it would be beneficial to include the other extreme in the form of a mutant incapable of producing pyocyanin.

11. The c-di-GMP-binding site mutants exhibited binding to DNA that was below that observed for the wild-type protein in the absence of c-di-GMP. Considering that these were not misfolded (L. 211), discuss what do you think accounts for the lack of DNA binding in the absence of c-di-GMP.

12. The authors demonstrate that the presence of pyocyanin enhances the BrIR-DNA binding capability. Does this mean that elevated levels of pyocyanin but low levels of c-di-GMP can render *P. aeruginosa* resistant to antimicrobial agents? While I understand that the authors provide gene expression data to support their claim, the authors should strongly consider testing the susceptibility of *P. aeruginosa* in vivo to address this question and further support their claim. This is even more important considering that binding by c-di-GMP and pyocyanin to BrIR are not having an additive effect, at least not with respect to DNA binding.

13. For EMSAs, loading controls and densitometry data should be provided.

14. Only EMSAs for BrIR variants harboring up to 4 substitutions in the c-di-GMP binding site are shown. Is DNA binding affected in BrIR variants harboring only 1 substitution in the c-di-GMP binding site? It might also be useful to include the *mexA* and *mexE* promoter regions, as was done for pyocyanin.

15. Rather than showing a structure of BrIR and pyocyanin based on modeling, it would be beneficial to show the structure based on the crystal structure. Are the same interactions with c-di-GMP and pyocyanin noted when BrIR crystals are soaked with the indicated molecules?

16. Along the same lines, the interaction between BrIR and pyocyanin is of concern. For one, the interaction appears to be based on hydrophobicity. And secondly, the EMSA data using BrIR variants of the pyocyanin binding residues are not entirely convincing.

17. The meaning of some of the additional supplemental files is unclear. If I understand these correctly, the structure of the C-terminal portion of the protein is not of high quality and thus, not as certain as the N-terminal portion of BrIR?!

Minor Comments

1. Intro is a bit disjointed lacking clear logical transitions or connections between c-di-GMP and pyocyanin. The link between c-di-GMP and pyocyanin and why you went after this needs to be clarified or revised in the introduction.

2. In the results section, the transition to pyocyanin seems a bit disjointed.

3. It would be beneficial to have a figure demonstrating the structures of the compounds that BrIR binds, including EtBr, c-di-GMP, pyocyanin, and 3A2P.
4. The authors claim that the BrIR-C domain is conserved (Line 229), but Supplementary Figure 3 shows that the degree of conservation is quite low.
5. Line 47: "which is a crucial factor for *P. aeruginosa* strains that chronically infect cystic fibrosis patients"
6. Line 52: "the pyocyanin receptor" instead of "pyocyanin receptor'.
7. Line 69: "similar to" "similarly to"
8. Is there functional similarity between c-di-GMP and pyocyanin? Explain, if so.
9. More Kd values for the binding interaction between c-di-GMP and its targets have been published by Chou and Galperin (J Bacteriol, 2016).
10. Line 169: "To understand how" instead of "to understand know how".
11. "Free DNA was more abundant than" instead of "The amount of free DNA was much more than the amount of bound DNA"
12. Line 226: "MerR family of multidrug transport activators" instead of "MerR family multidrug transport activators"
13. Line 245: "values" instead of "valules"
14. Line 267: "at the bottom" instead of "on the bottom"
15. Line 295: "The pyocyanin molecule is expected to be situated in the hydrophobic pocket" since this is a model.
16. Figure 4b: "BrIR variants" instead of 'BrIR varients"
17. Figure 4e: "Fold change in brIR expression" instead of "Relative fold of PbrIR"
18. Figure 5b: "BrIRY249A" instead of "BrIR249A"

Reviewer #2 (Remarks to the Author):

This manuscript by Wang et al. describes primarily the structures of the MerR-family member BrIR from *Pseudomonas aeruginosa*. Additional in vitro and in vivo studies are also reported in efforts to buttress the structural findings. BrIR is an important protein in this opportunistic pathogen, as it is involved in biofilm formation and multidrug resistance. These structures include a BrIR-(c-di-GMP) complex, the unliganded BrIR complex (the apo form), and the multidrug-binding C-terminal domain bound to a pyocyanin analogue. The structural studies are utilized to inform additional structure-guided mutagenesis studies and in vivo and in vitro functional studies. In general the structural studies are well done and the additional work is supportive. Although not the first BrIR-(c-di-GMP)

complex structure reported, the work does describe the first pyocyanin-receptor/effector complex as well as a ligand-free view of the BrIR complex. Comparison of the BrIR-(c-di-GMP) and apo BrIR complexes is somewhat informative as it points out the potential structural changes that this protein is able to make, i.e., it points out the potential conformational plasticity of BrIR that is necessary for its function. The structure also provides insight into the binding mechanism of pyocyanin, which is a reported activator of the brlR gene. In all this manuscript provides additional insight into the mechanisms of c-di-GMP and pyocyanin gene regulation. However, there are multiple issues that the authors must address.

The authors carry out c-di-GMP binding studies and report a $K_d = 2.69$ mM. The binding isotherm is fit with a single-site binding model. This affinity is approximately 1000 fold lower than that reported by the Sauer group (2014) and appears far beyond the concentration of c-di-GMP that might be found in *Pseudomonas aeruginosa*. The authors should definitely include a sentence that states the range of concentrations that c-di-GMP runs in this bacterium. Their current value seems very nonphysiologically relevant. Furthermore, the authors do not discuss the discrepancy between their finding and that of the Sauer group and must, especially given their statement on page 6, lines 164-167 that suggest that many additional c-di-GMP receptors might have such low affinity binding modes. Further, Wang et al. use SPR for their measurements and do not provide values for k_{on} and k_{off} . These should be given, as it appears that the k_{off} is very fast (Figure 2f). Also why do the authors use a single-site binding model when their crystal structure shows two binding sites. This suggests that either one site in their crystal structure is artefactual due to the concentrations used in the crystal set ups or their binding experiments are not reflective of the two-site binding mode of the c-di-GMP. The authors should consider an alternative mode to determine the K_d such as the use of a fluorescence polarisation-base assay as described in Tschowri et al. (2014) or DRaCALA described in Roelofs et al. (2009) Finally, the authors must address the different stoichiometries that they report and that the Sauer group reported. Is there an experimental reason for the 1:2 (c-di-GMP:BrIR subunit) value that Sauer reports and the 1:1 (c-di-GMP:BrIR subunit) value that the current authors report.

Page 6, paragraph 3: The authors should acknowledge that the coiled-coils of MerR have already been shown to show conformational plasticity including BmrR (Kumaraswami et al. 2009). More important to this discussion, is it possible that the conformational changes that the authors report for the apo and c-di-GMP are due to packing given that one form is I4 and the other form is P65? This should be discussed briefly.

The authors show the interactions between BrIR and c-di-GMP (shown in Figure 2a,2d,2e). They should include discussion explaining why BrIR binds c-di-GMP specifically and cannot bind c-di-AMP.

The structure of the BrIR-(c-di-GMP) complex shows an interesting tetrameric arrangement, which is also seen by Raju and Sharma. Further, both sets of authors carry out size exclusion chromatography (SEC) experiments to show that indeed this complex is a tetramer in solution. Given the unusual shape of the complex, which could easily result in unusual migration on the SEC column, the authors should have included a dimeric version of a MerR family member to allow the readers to see that elution profile. At the least the authors should have included a plot that showed log MW versus (Elution volume/Void volume) with a series of appropriate molecular weight markers for evaluation. This is necessary again as the Sauer group did not observe such behaviour and indeed found that c-di-GMP stabilised a dimeric form of BrIR over a monomeric form. These inconsistencies must be addressed.

One major item that the authors do not address directly impinges upon their models in Figure 7. That is the stoichiometry of DNA binding by BrIR. This can be evaluated by isothermal titration calorimetry or by fluorescence polarisation-based DNA binding assays. The result would readily distinguish

between the two. Also, the authors must include more information about their BrIR-(c-di-GMP) complex structure with particular emphasis on the distance of the recognition helices of the wHTH motif. Like most MerR family members, BrIR has an unusually long spacer (19 bp) between the -10 and -35 elements that require the protein-effector complex to distort the DNA significantly in order to effect transcription. What is the distance between the "recognition" helices that would bind consecutive major grooves of cognate DNA sites? Is this distance on par with those found in the DNA-bound forms of other MerR proteins? Is there evidence that BrIR does indeed undertwist/distort its DNA binding sites? Is DNA an active player in the final twisted conformation?

Page 9, paragraph 1: The authors suggest that tobramycin and gentamicin would bind in the same site, as do pyocyanin and ethidium. Do tobramycin and gentamicin compete with pyocyanin or ethidium for binding to the C-terminal domain of BrIR? A simple competition experiment as carried out in Figure 5b should provide insight into this important question.

Page 9, lines 257-258: This statement is a bit strong and should be toned down to something like, "This strong interaction suggested that pyocyanin is most probably a key physiologically relevant ligand for BrIR."

Page 9, lines 260-261: The BrIR-C domain-pyocyanin crystals do not diffract poorly because of the low solubility of pyocyanin. They diffract to low resolution due to poor packing or internal disorder or the liquid used to solubilise the pyocyanin or other reasons that might well be related to the incomplete occupancy of the compound. What was the resolution of these crystals?

Page 9, line 269: BrIR would not be the first multidrug binding transcription factor to be seen to bind more than one drug simultaneously. Although in the TetR family, QacR was shown to bind two drugs simultaneously (Schumacher et al., 2004). The principal is the same and should be noted.

Page 11, lines 302-303. The authors posit that 3A2P is too small as compared to pyocyanin to produce the necessary driving force to induce a conformational change upon binding BrIR. These molecules look very similar to this reviewer in terms of size. However, in terms of exocyclic atoms they are different and that might be the underlying reason why pyocyanin is an activator and 3A2P is not.

Figure 4a shows a series of gel shift assays. The improved binding of DNA in the presence of the BrIR-(c-di-GMP) complex is not impressive as there is a large amount of "free" DNA at the bottom of the gel. Is there a technical reason for this? Perhaps the experiment was not designed perfectly because the amount of c-di-GMP that is added is not 10 fold above the earlier reported K_d. Using 30 mM c-di-GMP would ensure that at least 90% of the BrIR would be bound to this signalling molecule.

Figure 6g: It is unclear how the authors quantified the results from the gel shift. Visual inspection of the gel shows very little difference between the wild type protein and the putative mutants.

Page 13, lines 378-381: The authors must provide at least one reference for these statements, especially the accumulation of high concentrations of antibiotics in biofilms.

Page 17, Gel Filtration Assay: This assay does not appear to be done properly with respect to the concentration of c-di-GMP included. The authors use 50 μ M c-di-GMP in the SEC buffer. According to their own binding assay and determined K_d (2.69 mM) for c-di-GMP, nearly all of the BrIR would not be bound by c-di-GMP.

Minor points:

Page 5, line 150: The phrase "terminal oxygen" is not the best description of the side chain hydroxyl group of tyrosine. This should be changed appropriately.

Figure 1h: the authors should use larger labels as they are now very difficult to see.

Figure 3c is nearly incomprehensible due to colour choice and the amount of material that is overlaid. The authors should include a stereoview of this overlay.

Table 1: the authors should place all unit cell lengths on one line and all unit cell angles on a separate line.

Reviewer #1 (Remarks to the Author):

The manuscript “BrlR from *Pseudomonas aeruginosa* is a common receptor for both cyclic di-GMP and pyocyanin” by Wang and colleagues examines potential binding partners for the transcriptional regulator BrlR. The authors use a structural approach to examine binding of the BrlR protein to c-di-GMP, including identifying specific amino acids that directly interact with c-di-GMP. After identifying two c-di-GMP binding sites, the authors then examine how these sites impact binding of BrlR to target promoter regions and find that both sites are required for optimal DNA binding. Unusually for a MerR-family protein, BrlR appears to form a tetramer in the absence/presence of effector molecules, with minimal structural changes noted upon binding c-di-GMP. The conformational changes do not appear to be sufficient to free up predicted DNA binding domains. In addition, data was presented indicating pyocyanin also binds to BrlR and affects the efficacy of BrlR:DNA interactions. The authors present a compelling, well-organized case for the importance of both c-di-GMP and pyocyanin in modulating the ability of BrlR to bind to DNA target regions. However, the study also raised several concerns as detailed below.

Comments:

1. Compared to other c-di-GMP binding proteins and the K_D for the *P. aeruginosa* BrlR reported in the literature, the K_D reported here is 1000-fold higher. The high K_D value was obtained using non-physiological conditions, as it would be unlikely that as much as 3 mM c-di-GMP would be present in a bacterial cell. Similarly, the K_D value for pyocyanin is also higher than expected (and may imply binding under non-physiological conditions as well). It would be useful to see if efficient binding could be obtained and studying using lower concentrations of potential effector molecules, or better expression why such high concentrations were utilized (see next comment).

AU: In order to obtain a more accurate K_D value, we performed the fluorescence polarization (FP)-based binding assays by using a fluoresceinated c-di-GMP analog 2'-Fluo-AHC-c-di-GMP. This time we obtained a new K_D of $7.33 \pm 0.53 \mu\text{M}$, which is in agreement with the K_D value that reported by Chambers et al¹. The competition assay shows 2'-Fluo-AHC-c-di-GMP is outcompeted completely by c-di-GMP from the binding sites (Supplementary Fig. 2), indicating that the K_D value is meaningful.

Our structure of BrlR in complex with c-di-GMP shows that c-di-GMP could be well replaced by c-di-AMP. The fluorescence polarization (FP)-based binding assay with 2'-Fluo-AHC-c-di-AMP gave a K_D of $12.94 \pm 0.65 \mu\text{M}$ (Fig. 2i). The fact that HTH domain of BrlR does recognize both c-di-GMP and c-di-AMP may have no physiological significance in this case since c-di-AMP does not exist in *P. aeruginosa*. However, for Gram-positive bacteria which contain c-di-AMP and mycobacteria which contain both c-di-AMP and c-di-GMP², the ability of MerR family protein to recognize both c-di-GMP and c-di-AMP may have significant physiological consequence.

The SPR results show that BrlR bound pyocyanin with a K_D of $4.52 \mu\text{M}$. This K_D may also seem a little bit higher than expected. However there are some biochemical data which show that this value does not deviate much from the physiological value. It has been observed that the production of pyocyanin by *pseudomonas aeruginosa* culture over one day is 1.5 to 17 μM in both minimal or rich medium³. This means pyocyanin in *P.aeruginosa* culture can reach a level comparable to the K_D we obtained. Very recently, the Per-Arnt-Sim (PAS) domain of

PA14_07500 (RmcA) was reported to be a pyocyanin receptor with a K_D of 3 μM by using the intrinsic fluorescence quenching method, which is also comparable to the value for BrIR⁴. Overall, our K_D value for pyocyanin lies in a reasonable range.

During the procedure for determining the binding affinity between BrIR and c-di-GMP we noticed the fluorophore of 2'-Fluo-AHC-c-di-GMP has a phenazine ring structure similar to procyanin (Fig. 2f). This raises a concern that whether the FP assays is interfered by the binding of BrIR and the fluorophore. The competition experiments with c-di-GMP (Supplementary Fig. 2) confirmed that both 2'-Fluo-AHC-c-di-GMP and c-di-GMP compete for the same binding sites. Furthermore, when we performed the same assay with BrIR-C domain which does not bind c-di-GMP, a K_D of $88.47 \pm 9.87 \mu\text{M}$ was obtained for the binding of BrIR-C to 2'-Fluo-AHC-c-di-GMP (Fig. 5d). This value is much higher than 7.33 μM . The big gap between these two K_D values and the result of the above-mentioned competition assay indicates that even the interaction between BrIR-C domain and the fluorophore causes some interferences, the effect should not be very significant. We add this part to the section of "The multidrug-binding domain of BrIR binds diverse toxic compounds" in the revised manuscript.

2. Is the high K_D value related to the method or the BrIR protein itself? It may be beneficial to compare the K_D of known c-di-GMP protein (other than BrIR) using the method described in the manuscript. If the method is solid, this raises some serious concerns about the BrIR protein and whether the variants produced in *E. coli* are the real native structure.

AU: The unexpectedly high K_D value obtained previously could be due to three reasons:

- (1) SPR immobilization: BrIR was immobilized on an activated CM5 sensor chip by amine coupling, which could result in complicated conditions. Each BrIR tetramer has four terminal amines and thus could be immobilized by one or two covalent bonds considering the steric hindrance (Fig. 1a). In addition, it has also been reported that nonspecific immobilization through one or more lysine residues can also result in an artificially reduced affinity⁵.
- (2) The nature of BrIR protein: Each BrIR monomer has two c-di-GMP binding sites, of which one is located at the N-terminus. Consequently, the covalent modification of the N-termini will cause a steric hindrance that affects c-di-GMP binding.
- (3) Oligomers of c-di-GMP: In order to detect the artificially reduced affinity, large amount of c-di-GMP was added in the solution. However, c-di-GMP forms heterogeneous oligomers at high concentration⁶, which means that the addition of large amount of c-di-GMP will only generate a very low effective concentration of free c-di-GMP molecules.

These would explain why SPR gave a very weak artificial affinity—an unexpectedly high K_D value. In the revised manuscript, we solve this problem by using fluorescence polarization (FP)-based binding assay with 2'-Fluo-AHC-c-di-AMP. The new K_D value is in the μM range, which is similar to the K_D value reported by Chambers *et al.*¹.

Meaning while, the pyocyanin binding site on BrIR-C domain is not affected by immobilization, so the K_D value for pyocyanin obtained using SPR assays is reliable. In our revised manuscript, we also employed the BrIR protein purified from *P. aeruginosa* PAO1 strain to conduct gel-filtration and EMSA assays, and that both BrIR proteins from *E. coli* and *P. aeruginosa* showed the same results (Fig. 1f).

3. The authors demonstrate that the apo-form of BrIR binds to DNA. This is a surprising finding that should warrant further testing as it may indicate lack of specificity. Therefore, I strongly suggest testing BrIR DNA binding using control promoters (e.g. those believed not to be linked to BrIR).

AU: Thanks for the suggestion. We conducted the EMSA experiment with the promoter of P_{pscE-F} that is not linked to BrIR¹ as a negative control, and the results confirmed our previous findings (new Fig. 4a). In the other lanes, 5 ng/mL poly[dI-dC] was added as a non-specific competitor against P_{brIR} for BrIR binding.

4. The specificity of binding to BrIR could be clarified by testing additional compounds related to c-di-GMP and pyocyanin. For example, cAMP could be tested to see if BrIR binds to cyclic nucleotides in general or c-di-GMP specifically. Likewise, other non-pyocyanin phenazines could determine BrIR specificity for pyocyanin.

AU: Thanks for the suggestions. We have tested c-di-AMP binding in FP binding assays. The results showed that BrIR also binds c-di-AMP with a K_D of $12.94 \pm 0.65 \mu\text{M}$ (Fig. 2i). The structural analysis indicated that both c-di-GMP and c-di-AMP fit the two binding sites on BrIR (Supplementary Fig. 3a, b). We also found that BrIR does not bind AMP in SPR experiments. Our result is in line with the previous work which proved that binding of c-di-GMP to BrIR is not affected by GTP or cAMP¹. That means cAMP does not bind to BrIR specifically. Then, BrIR may bind to cyclic-di-nucleotides in general although c-di-AMP does not exist in *Pseudomonas aeruginosa*.

As mentioned above, we found BrIR-C domain binds to fluorophore of 2'-Fluo-AHC-c-di-GMP (Fig. 5d). We also determined the structure of BrIR-C in complex with the pyocyanin analogue 3A2P (Fig. 5a,f). Thus it is reasonable to speculate that other non-pyocyanin phenazines are also able to bind to BrIR. First, we performed molecular docking with pyocyanin and other pyocyanin analogs. The results showed that the rigid groove of BrIR-C could bind all pyocyanin-like molecules (Fig. 6f). However, their exact binding sites are different. Pyocyanin is the only compound that slides toward the HTH motif during the docking process, while phenazine-1-carboxylic acid (PCA) and phenazine-1-carboxamide (PCN) are expected to be located at the same position as 3A2P (Fig. 6e). 3A2P has no interaction with the other HTH domain of BrIR, thus does not enhance the DNA-binding of BrIR. Therefore, the other non-pyocyanin phenazines cannot activate BrIR unless they bind to the groove of BrIR-C domain like what pyocyanin does. Second, we conducted the RT-PCR experiments. qPCR results showed that there is a 4-fold enrichment of *PbrIR* in PAO1 than in a PAO1: $\Delta phzA1-G1/phzA2-G2$ strain, which does not produce any phenazine molecules⁷. Furthermore, there is also a 5.3-fold enrichment of *PbrIR* in PAO1 than in a PAO1: $\Delta phzMSH$ strain, which still produces PCA but not produce pyocyanin and other pyocyanin analogs⁷. These results suggested that PCA cannot increase the transcription level of *brIR*. Therefore, pyocyanin is an excellent transcriptional activator of *brIR* both in vivo and in vitro.

5. The hypothetical models for the c-di-GMP-responsive binding to BrIR seem to be rather speculative. It would be useful to perform additional experiments to elucidate the response to c-di-GMP binding or eliminate them entirely.

AU: The hypothetical models (previous Fig. 7) and related statements were deleted in the current version of manuscript.

6. The authors indicate that BrIR, unlike other MerR proteins, forms tetramers (dimers of dimers). Given the unusual oligomerization status of BrIR and to ensure that this is not an artifact due to protein overproduction in *E. coli*, the authors should consider determining whether the tetramer is detectable in *P. aeruginosa*.

AU: We purified BrIR protein expressed in *P. aeruginosa* PAO1 strain harboring a pHERD20T-BrIR vector. Gel-filtration assay showed that BrIR protein from *P. aeruginosa* has the same elution volume as that of the BrIR protein expressed and purified from *E. coli*, which suggested that BrIR proteins from both *P. aeruginosa* and *E. coli* have similar molecular sizes and should be tetramers (new Fig. 1f).

7. It would be interesting to see if BrIR could be crystalized in the presence of both c-di-GMP and the pyocyanin analogue 3A2P. This would provide valuable information about potential steric hindrance caused by binding to one ligand versus another.

AU: We totally agree with this reviewer that the ternary complex structure of BrIR with c-di-GMP and 3A2P will provide more insights about the interactions among these three molecules. At first, we tried the co-crystallization of BrIR with c-di-GMP and 3A2P, but failed to produce any crystal. Then we tried soaking methods. After solving several structures of BrIR/c-di-GMP crystal soaked in the buffer containing various concentrations of 3A2P, no ternary complex was observed. We are still trying to make a breakthrough in this task, and would like to include any new observations in our future report.

8. The mutants Y183A and Y249A are said to exhibit a weak response to pyocyanin (L. 311-313), yet Fig.6 seems to suggest that the response is the same as for the wild-type protein.

AU: BrIR-C domain, which is a GyrI-like domain (PF06445), has a rigid structure and provides a versatile platform for diverse ligands' binding as its homologous proteins do⁸. A single mutation in the ligand-binding pocket, which does not influence the HTH motif and the conformation of BrIR, should not affect DNA binding unless pyocyanin exists. It has been reported mutation of Glu-253 to either alanine (E253A) or glutamine (E253Q) eliminates ligand binding of BmrR-C domain, a homologue of BrIR-C domain⁹. The structural analysis of BrIR-C domain showed that Glu247 should be the key residue which plays the same role in ligand binding as E253 does in BmrR-C domain (Fig. 5a). To test the role of Glu247, we constructed three BrIR mutants E247A, E247Q, and E247Y. However, all these three mutants are insoluble. We also tried to mutate other nearby residues in which F253R is not expressed, and C173W has a poor behavior. Finally, two residues which do not interact with pyocyanin but stabilize E247A were mutated to Alanine to see if we

could get some valuable data. Notably, according to the EMSA results, wild-type BrlR bind 5.47-fold of more DNA in the presence of pyocyanin than in the absence pyocyanin. For Y183A and Y249A the fold numbers are 3.23 and 4.15 respectively, which are smaller than 5.47 (Fig. 6g). So we mentioned they have a slightly weaker response to pyocyanin. We agree that these data are not decisive. But we have tried our best and cannot get better results.

9. It would be beneficial to expand the qPCR experiment regarding DGC PA4843 overexpression to include the mutant BrlR variants, or to repeat it with the Miller assay strains.

AU: We have repeated β -galactosidase assays for the BrlR variants with DGC PA4843 overexpression in the current manuscript.

10. While conditions correlating with higher pyocyanin production were presented, it would be beneficial to include the other extreme in the form of a mutant incapable of producing pyocyanin.

AU: We did the RT-PCR experiments in PAO1: Δ *phzA1-G1/phzA2-G2* and PAO1: Δ *phzMSH* strains that are deficient to produce pyocyanin. Interestingly, significant reduction in the transcriptional level of *PbrlR* was observed in both mutant strains relative to that in wild-type strain. This data highlight the specificity of BrlR for pyocyanin in vivo (please see no. 4 above).

11. The c-di-GMP-binding site mutants exhibited binding to DNA that was below that observed for the wild-type protein in the absence of c-di-GMP. Considering that these were not misfolded (L. 211), discuss what do you think accounts for the lack of DNA binding in the absence of c-di-GMP.

AU: Two c-di-GMP-binding sites are located in the HTH motif of BrlR. Several arginine residues play critical roles in c-di-GMP binding (Fig. 3c,d). The mutations of these residues would significantly reduce positive charge on the surface of the HTH motif, thus lead to weaker DNA binding compare to wild-type BrlR.

12. The authors demonstrate that the presence of pyocyanin enhances the BrlR-DNA binding capability. Does this mean that elevated levels of pyocyanin but low levels of c-di-GMP can render *P. aeruginosa* resistant to antimicrobial agents? While I understand that the authors provide gene expression data to support their claim, the authors should strongly consider testing the susceptibility of *P. aeruginosa* in vivo to address this question and further support their claim. This is even more important considering that binding by c-di-GMP and pyocyanin to BrlR are not having an additive effect, at least not with respect to DNA binding.

AU: We thank the reviewer for pointing out this important issue. We tested the susceptibility of planktonic *P. aeruginosa* cells to tobramycin treatment using the protocol adapted from the methods previously described by Sauer's group¹⁰. We found that the addition of pyocyanin renders planktonic wild-type PAO1 cells but not Δ *brlR* cells resistant to tobramycin. Interestingly, for the cells harboring pHERD20T-PA2133, elevated level of pyocyanin is still able to increase their tobramycin resistance (new Fig. 7a). We discussed these results in Page 15, and suggested that the

elevated level of pyocyanin renders *P. aeruginosa* resistant to tobramycin via a *brlR*-dependent mechanism, which possibly bypasses the regulation of c-di-GMP level.

13. For EMSAs, loading controls and densitometry data should be provided.

AU: Loading controls and densitometry data have been provided in the new Fig. 4a.

14. Only EMSAs for BrlR variants harboring up to 4 substitutions in the c-di-GMP binding site are shown. Is DNA binding affected in BrlR variants harboring only 1 substitution in the c-di-GMP binding site? It might also be useful to include the *mexA* and *mexE* promoter regions, as was done for pyocyanin.

AU: EMSA experiments including the *mexA* and *mexE* promoter regions in the presence of c-di-GMP have been performed as suggested (see Fig. 4d,e).

15. Rather than showing a structure of BrlR and pyocyanin based on modeling, it would be beneficial to show the structure based on the crystal structure. Are the same interactions with c-di-GMP and pyocyanin noted when BrlR crystals are soaked with the indicated molecules?

AU: We agree with this reviewer. In order to achieve this goal we have performed numerous co-crystallization and soaking experiments. However, high concentrations of pyocyanin always make BrlR or BrlR-C domain crystal lose diffraction, no matter by using co-crystallization or soaking method. When BrlR/c-di-GMP crystals were soaked with low concentrations of pyocyanin or pyocyanin analog 3A2P, no pyocyanin or 3A2P was observed in the solved structure. In the end, we got the structure of BrlR-C in complex with a water-soluble pyocyanin analog 3A2P by co-crystallization. 3A2P and pyocyanin have the same phenazine ring. The molecular docking results of pyocyanin based on the structure of BrlR-C/3A2P are rather convincing, which are consistent with our EMSA results (Fig. 6a,f).

16. Along the same lines, the interaction between BrlR and pyocyanin is of concern. For one, the interaction appears to be based on hydrophobicity. And secondly, the EMSA data using BrlR variants of the pyocyanin binding residues are not entirely convincing.

AU: The problem is we cannot get a soluble mutant protein by modifying the critical residue E247. The detail of this problem has been discussed in No.8.

17. The meaning of some of the additional supplemental files is unclear. If I understand these correctly, the structure of the C-terminal portion of the protein is not of high quality and thus, not as certain as the N-terminal portion of BrlR?!

AU: We revised supplemental files, and hope this version would be clear. The crystal structure of the BrlR-C domain was determined at a high resolution of 1.4 Å, and the data of both termini is very reliable. Sorry for not making this clear in the old version of text.

Minor Comments

1. Intro is a bit disjointed lacking clear logical transitions or connections between c-di-GMP and pyocyanin. The link between c-di-GMP and pyocyanin and why you went after this needs to be clarified or revised in the introduction.

AU: Thanks for the suggestion. We add one paragraph in the introduction (page 2) to clarify the link between c-di-GMP and pyocyanin, which would help the logical transition from c-di-GMP to pyocyanin.

2. In the results section, the transition to pyocyanin seems a bit disjointed.

AU: We added some sentences to make the transition from multidrug-binding to pyocyanin binding in the results (page 10) part.

3. It would be beneficial to have a figure demonstrating the structures of the compounds that BrlR binds, including EtBr, c-di-GMP, pyocyanin, and 3A2P.

AU: Structures of the compounds that BrlR binds are shown in the new Fig. 5b.

4. The authors claim that the BrlR-C domain is conserved (Line 229), but Supplementary Figure 3 shows that the degree of conservation is quite low.

AU: The other similar GyrI-like proteins of BrlR-C domain in the new Supplementary Fig. 4 were generated by a DALI search¹¹. Pairwise structural superimpositions reveal a mean rmsd values from 2.8 Å to 3.8 Å. Thus the structure of the BrlR-C domain is conserved among MerR-family members (Supplementary Fig. 1). Although their amino acid sequence identity is low, the key residues in the consensus sequence are >80% conserved (Supplementary Fig. 4), and similar observations have been reported by Herschel Wade's group⁸.

5. Line 47: "which is a crucial factor for *P. aeruginosa* strains that chronically infect cystic fibrosis patients"

AU: This sentence was deleted.

6. Line 52: "the pyocyanin receptor" instead of "pyocyanin receptor".

AU: A new paragraph about pyocyanin was added.

7. Line 69: "similar to" "similarly to"

AU: It has been corrected as suggested.

8. Is there functional similarity between c-di-GMP and pyocyanin? Explain, if so.

AU: The functional similarity between c-di-GMP and pyocyanin has been explained in the introduction. 'It has been reported that pyocyanin facilitates biofilm formation by promoting extracellular DNA release¹²⁻¹⁴ and that its degradation by the demethylase PodA disrupts *P. aeruginosa* biofilms. Just as c-di-GMP, pyocyanin also enhances resistance of *P. aeruginosa* to many antimicrobial reagents by up-regulating the expression of multidrug efflux pumps'.

9. More Kd values for the binding interaction between c-di-GMP and its targets have been published by Chou and Galperin (J Bacteriol, 2016).

AU: The reference was added in the text as suggested (Reference 27).

10. Line 169: "To understand how" instead of "to understand know how".

AU: Corrected.

11. "Free DNA was more abundant than" instead of "The amount of free DNA was much more than the amount of bound DNA"

AU: This sentence was revised as suggested.

12. Line 226: "MerR family of multidrug transport activators" instead of "MerR family multidrug transport activators"

AU: This sentence was revised.

13. Line 245: "values" instead of "valules"

AU: Corrected.

14. Line 267: "at the bottom" instead of "on the bottom"

AU: Corrected.

15. Line 295: "The pyocyanin molecule is expected to be situated in the hydrophobic pocket" since this is a model.

AU: Revised as suggested.

16. Figure 4b: "BrlR variants" instead of 'BrlR varients"

AU: Corrected as suggested in the new Figure 4c-e.

17. Figure 4e: "Fold change in brlR expression" instead of "Relative fold of PbrlR"

AU: Revised as suggested in the new Fig. 4h.

18. Figure 5b: “BrlRY249A” instead of “BrlR249A”

AU: It has been corrected as suggested in the new Fig. 5c.

Reviewer #2 (Remarks to the Author):

This manuscript by Wang et al. describes primarily the structures of the MerR-family member BrlR from *Pseudomonas aeruginosa*. Additional in vitro and in vivo studies are also reported in efforts to buttress the structural findings. BrlR is an important protein in this opportunistic pathogen, as it is involved in biofilm formation and multidrug resistance. These structures include a BrlR-(c-di-GMP) complex, the unliganded BrlR complex (the apo form), and the multidrug-binding C-terminal domain bound to a pyocyanin analogue. The structural studies are utilised to inform additional structure guided mutagenesis studies and in vivo and in vitro functional studies. In general the structural studies are well done and the additional work is supportive. Although not the first BrlR-(c-di-GMP) complex structure reported, the work does describe the first pyocyanin-receptor/effector complex as well as a ligand-free view of the BrlR complex. Comparison of the BrlR-(c-di-GMP) and apo BrlR complexes is somewhat informative as it points out the potential structural changes that this protein is able to make, i.e., it points out the potential conformational plasticity of BrlR that is necessary for its function. The structure also provides insight into the binding mechanism of pyocyanin, which is a reported activator of the brlR gene. In all this manuscript provides additional insight into the mechanisms of c-di-GMP and pyocyanin gene regulation. However, there are multiple issues that the authors must address.

The authors carry out c-di-GMP binding studies and report a $K_d = 2.69$ mM. The binding isotherm is fit with a single-site binding model. This affinity is approximately 1000 fold lower than that reported by the Sauer group (2014) and appears far beyond the concentration of c-di-GMP that might be found in *Pseudomonas aeruginosa*. The authors should definitely include a sentence that states the range of concentrations that c-di-GMP runs in this bacterium. Their current value seems very nonphysiologically relevant. Furthermore, the authors do not discuss the discrepancy between their finding and that of the Sauer group and must, especially given their statement on page 6, lines 164-167 that suggest that many additional c-di-GMP receptors might have such low affinity binding modes. Further, Wang et al. use SPR for their measurements and do not provide values for k_{on} and k_{off} . These should be given, as it appears that the k_{off} is very fast (Figure 2f). Also why do the authors use a single-site binding model when their crystal structure shows two binding sites. This suggests that either one site in their crystal structure is artefactual due to the concentrations used in the crystal set ups or their binding experiments are not reflective of the two-site binding mode of the c-di-GMP. The authors should consider an alternative mode to determine the K_d such as the use of a fluorescence polarisation-based assay as described in Tschowri et al. (2014) or DRaCALA described in Roelofs et al. (2009) Finally, the authors must address the different stoichiometries that they report and that the Sauer group reported. Is there an experimental reason for the 1:2 (c-di-GMP:BrlR subunit) value that Sauer reports and the 1:1 (c-di-GMP:BrlR subunit) value that the current authors report.

AU: The unexpectedly high K_D value obtained previously by the combination of the SPR method could be due to three reasons:

- (4) SPR immobilization: BrlR was immobilized on an activated CM5 sensor chip by amine coupling, which could result in complicated conditions. Each BrlR tetramer has four terminal amines and thus could be immobilized by one or two covalent bonds considering the steric hindrance (Fig. 1a). In addition, it has also been reported that nonspecific immobilization through one or more lysine residues can also result in an artificially reduced affinity⁵.
- (5) The nature of BrlR protein: Each BrlR monomer has two c-di-GMP binding sites, of which one is located at the N-terminus. Consequently, the covalent modification of the N-termini will cause a steric hindrance that affects c-di-GMP binding.
- (6) Oligomers of c-di-GMP: In order to detect the artificially reduced affinity, large amount of c-di-GMP was added in the solution. However, c-di-GMP forms heterogeneous oligomers at high concentration⁶, which means that the addition of large amount of c-di-GMP will only generate a very low effective concentration of free c-di-GMP molecules.

These would explain why SPR gave a very weak artificial affinity—an unexpectedly high K_D value. In the revised manuscript, we solve this problem by using fluorescence polarization (FP)-based binding assay with 2'-Fluo-AHC-c-di-AMP. The new K_D value is in the μM range, which is similar to the K_D value reported by Chambers *et al.*¹.

We also used a FP assays to determine the binding stoichiometry of c-di-GMP to BrlR. 1 nM F-c-di-GMP mixed with 160 μM c-di-GMP (which is 20-fold higher than the K_D of BrlR bound F-c-di-GMP) was incubated with increasing amount of BrlR. The graph of the resulting data shows a linear increase in the observed millipolarization until saturation of the c-di-GMP by BrlR, after which the millipolarization values showed no increase (Fig. 2h). The inflection point occurs at a BrlR concentration of $\sim 88 \mu\text{M}$, which, when divided by the concentration of c-di-GMP (160 μM), indicates a stoichiometry of one BrlR subunit to two c-di-GMP molecules, consistent with the structural data (Fig. 2a). It seems that our stoichiometric ratio is different than the Sauer group reported. However, we think there is an experimental reason for the 1:2 (c-di-GMP:BrlR subunit) value in the Sauer group. BrlR is tetramer with 8 c-di-GMP binding sites in solution. Then one or two c-di-GMP can bind four BrlR subunits in their pulldown assay. Hence, their binding ratio of c-d-GMP:BrlR subunit is lower than the actual binding ratio.

Page 6, paragraph 3: The authors should acknowledge that the coiled-coils of MerR have already been shown to show conformational plasticity including BmrR (Kumaraswami *et al.* 2009). More important to this discussion, is it possible that the conformational changes that the authors report for the apo and c-di-GMP are due to packing given that one form is I4 and the other form is P65? This should be discussed briefly.

AU: Crystal packing analyses of the apo-BrlR structure and the BrlR-c-di-GMP structure have been appended in the new supplementary Fig. 5 in the manuscript. Crystal packing analyses showed that the c-di-GMP molecules are the connectors between the neighboring tetramers in the BrlR-c-di-GMP structure. The long coiled-coils linker has no interaction with the protomers of adjacent tetramers. The direct interactions between the symmetric molecules in the apo-BrlR seem more extensive than that in the BrlR-c-di-GMP structure (supplementary Fig. 5a,b). It is unlikely the conformational changes of BrlR are due to different crystal packing.

The authors show the interactions between BrIR and c-di-GMP (shown in Figure 2a,2d,2e). They should include discussion explaining why BrIR binds c-di-GMP specifically and cannot bind c-di-AMP.

AU: Thanks for the suggestion. In our BrIR/c-di-GMP complex structure, BrIR mainly binds c-di-GMP through interactions with the phosphate group and the purine group (Fig. 2d,e). When we tentatively placed a c-di-AMP in the same place, the interactions between BrIR and the ligand in the two binding sites largely remain the same (Supplementary Fig. 3a,b). FP binding assays that used the similar fluoresceinated c-di-AMP analog (2'-fluo-AHC-c-di-AMP) revealed a K_D of $12.94 \pm 0.65 \mu\text{M}$ by BrIR (Fig. 2i), which is slightly larger than the K_D value of F-c-di-GMP. Thus, BrIR recognizes both c-di-GMP and c-di-AMP. The EMSA results showed that the addition of c-di-GMP or c-di-AMP both resulted in increased BrIR-DNA binding in a concentration-dependent manner (Fig. 4b). The fact that HTH domain of BrIR does recognize both c-di-GMP and c-di-AMP may have no physiological significance in this case since c-di-AMP does not exist in *P. aeruginosa*. However, for Gram-positive bacteria which contain c-di-AMP and mycobacteria which contain both c-di-AMP and c-di-GMP², the ability of MerR family protein to recognize both c-di-GMP and c-di-AMP may have significant physiological consequence.

The structure of the BrIR-(c-di-GMP) complex shows an interesting tetrameric arrangement, which is also seen by Raju and Sharma. Further, both sets of authors carry out size exclusion chromatography (SEC) experiments to show that indeed this complex is a tetramer in solution. Given the unusual shape of the complex, which could easily result in unusual migration on the SEC column, the authors should have included a dimeric version of a MerR family member to allow the readers to see that elution profile. At the least the authors should have included a plot that showed log MW versus (Elution volume/Void volume) with a series of appropriate molecular weight markers for evaluation. This is necessary again as the Sauer group did not observe such behaviour and indeed found that c-di-GMP stabilised a dimeric form of BrIR over a monomeric form. These inconsistencies must be addressed.

AU: We carried out size exclusion chromatography (SEC) experiments with BmrR as a control as suggested (Fig. 1f), and a plot to show log MW is also included in Fig. 1f. The SEC results showed that BmrR is a dimer, and BrIR is a tetramer with or without c-di-GMP or pyocyanin (Fig. 1f). We also purified BrIR from PAO1, and found that it is also a tetramer. In addition, we performed cross-linking assays with BrIR in solution. In this assay BrIR was incubated with di-(N-succinimidyl) ethylene glycol disuccinate (EGS) or dithiobis(succinimidyl propionate) (DSP) at room temperature. The SDS PAGE shows that there are two major protein bands which represent monomer and dimer respectively. Clear bands for tetramer or higher oligomer were not observed (Figure was shown in Supplementary Figure for Referees). This result is in agreement with our structure that suggests BrIR tetramer is a dimer of dimer. We speculate that the two monomers that compose a dimer (AD or BC) are able to be cross-linked. However the two monomers that do not belong to the same dimer are resistant to cross-linking due to stereochemistry (Fig. 1a).

One major item that the authors do not address directly impinges upon their models in Figure 7. That is the stoichiometry of DNA binding by BrIR. This can be evaluated by isothermal titration calorimetry or by fluorescence polarisation-based DNA binding assays. The result would readily distinguish between the two. Also, the authors must include more information about their BrIR-(c-di-GMP) complex structure with particular emphasis on the distance of the recognition helices of the wHTH motif. Like most MerR family members, BrIR has an unusually long spacer (19 bp) between the -10 and -35 elements that require the protein-effector complex to distort the DNA significantly in order to effect transcription. What is the distance between the “recognition” helices that would bind consecutive major grooves of cognate DNA sites? Is this distance on par with those found in the DNA-bound forms of other MerR proteins? Is there evidence that BrIR does indeed undertwist/distort its DNA binding sites? Is DNA an active player in the final twisted conformation?

AU: We had used FP and ITC experiments to determine the stoichiometry of DNA binding by BrIR. However, the binding of BrIR to 36bp PbrIR is weak, the molar ratio of BrIR: *PbrIR* was 600:1 in our EMSAs. Thus, we cannot detect the stoichiometry of DNA binding by BrIR easily. As a MerR family transcriptional activator, BrIR should be a dimer for one promoter DNA. Hence, we made the binding mode in previous Fig. 7. The Reviewer #1 also thought previous Fig.7 is hypothetical. Thus we deleted previous Fig. 7, and focused on the conformational change of the wHTH motif upon c-di-GMP binding in the discussion. A BrIR tetramer consists of a pair of MerR-likier dimer. Each dimer should recognize and bind one promoter DNA. The distance between two recognition helix ($\alpha 2$) of the HTH motif in BrIR dimer is much longer than that of BmrR dimer. After BrIR binds c-di-GMP, the distance ($C^\alpha(H20)-C^\alpha(H20')$) shortens by 1.2 Å (Supplementary Fig. 7a). This could possibly make BrIR recognize DNA easier. Based on various BmrR structures, we remodeled a complex structure of the HTH motif of BrIR with a double-stranded DNA. It seems that two c-di-GMP may contribute the DNA binding of BrIR (Supplementary Fig. 7c). The bound DNA of BrIR will occupy the wHTH motif and push the multidrug-binding domains of the other dimer away (Supplementary Fig. 7d). There are even some possibilities that the bound DNA induces the BrIR tetramer to split into two dimers. Then BrIR could undergo huge conformational changes to form a classical MerR-like dimer as BmrR-DNA complex. However without the crystal structure of BrIR in complex with DNA we cannot give a detailed mechanism.

Page 9, paragraph 1: The authors suggest that tobramycin and gentamicin would bind in the same site, as do pyocyanin and ethidium. Do tobramycin and gentamicin compete with pyocyanin or ethidium for binding to the C-terminal domain of BrIR? A simple competition experiment as carried out in Figure 5b should provide insight into this important question.

AU: We found that BrIR-C binds to the fluorophore of F-c-di-GMP. Then we performed the competition experiments by using FP assays with BrIR-C and F-c-di-GMP (Fig. 5e). The results show that tobramycin, the fluorophore (like the backbone of RH6G), and ethidium compete for the same site in BrIR-C.

Page 9, lines 257-258: This statement is a bit strong and should be toned down to something like,

“This strong interaction suggested that pyocyanin is most probably a key physiologically relevant ligand for BrlR.”

AU: We have detected BrlR binds F-c-di-GMP with a K_D of $7.33 \pm 0.53 \mu\text{M}$ (Fig. 2g). Hence, BrlR binds c-di-GMP and pyocyanin efficiently, so we deleted this sentence in the revision.

Page 9, lines 260-261: The BrlR-C domain-pyocyanin crystals do not diffract poorly because of the low solubility of pyocyanin. They diffract to low resolution due to poor packing or internal disorder or the liquid used to solubilise the pyocyanin or other reasons that might well be related to the incomplete occupancy of the compound. What was the resolution of these crystals?

AU: We failed to get structure of BrlR or BrlR-C in complex with pyocyanin through co-crystallization method. We then tried to get the complex structure by soaking BrlR/c-di-GMP crystal in a variety of concentrations of pyocyanin. At low concentrations of pyocyanin the diffraction of the crystals are reduced to 2.7-3.0 Å, and no pyocyanin was observed in the solved structure. At high concentrations of pyocyanin, however, the crystals have no diffraction at all. The structures of apo BrlR that we solved all have low resolution (3.2-5 Å), which is difficult for soaking pyocyanin or its analogs. The BrlR-C structure has high resolution (1.4 Å), but we cannot obtain the structure of BrlR-C in complex with pyocyanin by soaking method.

Page 9, line 269: BrlR would not be the first multidrug binding transcription factor to be seen to bind more than one drug simultaneously. Although in the TetR family, QacR was shown to bind two drugs simultaneously (Schumacher et al., 2004). The principal is the same and should be noted.

AU: As suggested, we have added this reference and rewritten the sentence (page 11).

Page 11, lines 302-303. The authors posit that 3A2P is too small as compared to pyocyanin to produce the necessary driving force to induce a conformational change upon binding BrlR. These molecules look very similar to this reviewer in terms of size. However, in terms of exocyclic atoms they are different and that might be the underlying reason why pyocyanin is an activator and 3A2P is not.

AU: We totally agree with this point. Both 3A2P and pyocyanin are small in size. Due to the difference in exocyclic atoms, they bind to BrlR-C domain in different positions. The small size of 3A2P means that it cannot make contact with the neighboring HTH domain at the current position. However, pyocyanin is expected to be close to the neighboring HTH domain, thus pushes against the R9 and E12 side chains of the HTH motif even it is also small in size (Fig. 6e). In general, the difference in the exocyclic atoms is the underlying reason.

Figure 4a shows a series of gel shift assays. The improved binding of DNA in the presence of the BrlR-(c-di-GMP) complex is not impressive as there is a large amount of “free” DNA at the bottom of the gel. Is there a technical reason for this? Perhaps the experiment was not designed perfectly because the amount of c-di-GMP that is added is not 10 fold above the earlier reported

Kd. Using 30 mM c-di-GMP would ensure that at least 90% of the BrIR would be bound to this signalling molecule.

AU: Thanks for the suggestion. We repeated the EMSA experiment with new design. By reducing the amount of DNA and adding more BrIR proteins, we got better results (Fig. 4a,b). The free DNA was reduced significantly in the present of 1.6 mM c-di-GMP.

Figure 6g: It is unclear how the authors quantified the results from the gel shift. Visual inspection of the gel shows very little difference between the wild type protein and the putative mutants.

AU: BrIR-C domain, which is a GyrI-like domain (PF06445), has a rigid structure and provides a versatile platform for diverse ligands binding as its homologous proteins do⁸. A single mutation in the ligand-binding pocket, which does not influence the HTH motif and the conformation of BrIR, should not affect DNA binding unless pyocyanin exists. It has been reported mutation of Glu-253 to either alanine (E253A) or glutamine (E253Q) eliminates ligand binding of BmrR-C domain, a homologue of BrIR-C domain⁹. The structural analysis of BrIR-C domain showed that Glu247 should be the key residue which plays the same role in ligand binding as E253 does in BmrR-C domain (Fig. 5a). To test the role of Glu247 we constructed three BrIR mutants E247A, E247Q, and E247Y. However, all these three mutants are insoluble. We also tried to mutate other nearby residues in which F253R is not expressed, and C173W has a poor behavior. Finally, two residues which do not interact with pyocyanin but stabilize E247A were mutated to Alanine to see if we could get some valuable data. Notably, according to the EMSA results, wild-type BrIR bind 5.47 folds more DNA in the presence of pyocyanin than in the absence pyocyanin. For Y183A and Y249A the fold numbers are 3.23 and 4.15 respectively, which are smaller than 5.47 (Fig. 6g). So we mentioned they have a slightly weaker response to pyocyanin. We agree that these data are not decisive. But we have tried our best and cannot get better results.

Page 13, lines 378-381: The authors must provide at least one reference for these statements, especially the accumulation of high concentrations of antibiotics in biofilms.

AU: Biofilm acts as a diffusion barrier for antibiotics¹⁵, thereby both diffusions of antibiotics into and outward biofilms are difficult. Consequently, the concentration of antibiotics in the biofilms of the antibiotics producing strains will accumulate to high concentrations¹⁶. The references have been appended.

Page 17, Gel Filtration Assay: This assay does not appear to be done properly with respect to the concentration of c-di-GMP included. The authors use 50 μ M c-di-GMP in the SEC buffer. According to their own binding assay and determined Kd (2.69 mM) for c-di-GMP, nearly all of the BrIR would not be bound by c-di-GMP.

AU: We have determined a new K_D ($7.33 \pm 0.53 \mu$ M) for c-di-GMP, therefore the c-di-GMP concentration used in gel filtration assay becomes reasonable. When we performed the gel filtration assay for BrIR with c-di-GMP, we design the assay based the K_D value reported by the Sauer group¹.

Minor points:

Page 5, line 150: The phrase “terminal oxygen” is not the best description of the side chain hydroxyl group of tyrosine. This should be changed appropriately.

AU: As suggested, we have revised this phrase.

Figure 1h: the authors should use larger labels as they are now very difficult to see.

AU: Fig. 1h has been recreated with larger labels.

Figure 3c is nearly incomprehensible due to colour choice and the amount of material that is overlaid. The authors should include a stereoview of this overlay.

AU: Fig. 3 has been recreated, and a stereoview is included.

Table 1: the authors should place all unit cell lengths on one line and all unit cell angles on a separate line.

AU: Table 1 has been rearranged as suggested.

Reference for the response letter:

- 1 Chambers, J. R., Liao, J., Schurr, M. J. & Sauer, K. BrlR from *Pseudomonas aeruginosa* is a c-di-GMP-responsive transcription factor. *Mol. Microbiol.* **92**, 471-487, (2014).
- 2 Romling, U. Great Times for Small Molecules: c-di-AMP, a Second Messenger Candidate in Bacteria and Archaea. *Science Signaling* **1**, (2008).
- 3 Sismaet, H. J., Webster, T. A. & Goluch, E. D. Up-regulating pyocyanin production by amino acid addition for early electrochemical identification of *Pseudomonas aeruginosa*. *Analyst* **139**, 4241-4246, (2014).
- 4 Okegbe, C. *et al.* Electron-shuttling antibiotics structure bacterial communities by modulating cellular levels of c-di-GMP. *Proc. Natl Acad. Sci. USA* **114**, E5236-E5245, (2017).
- 5 Kortt, A. A., Oddie, G. W., Iliades, P., Gruen, L. C. & Hudson, P. J. Nonspecific amine immobilization of ligand can be a potential source of error in BIAcore binding experiments and may reduce binding affinities. *Anal. Biochem.* **253**, 103-111, (1997).
- 6 Gentner, M., Allan, M. G., Zaehring, F., Schirmer, T. & Grzesiek, S. Oligomer Formation of the Bacterial Second Messenger c-di-GMP: Reaction Rates and Equilibrium Constants Indicate a Monomeric State at Physiological Concentrations. *J. Am. Chem. Soc.* **134**, 1019-1029 (2012).
- 7 Das, T. & Manefield, M. Phenazine production enhances extracellular DNA release via hydrogen peroxide generation in *Pseudomonas aeruginosa*. *Commun. Integr. Biol.* **6**, e23570, (2013).
- 8 Moreno, A. *et al.* Solution Binding and Structural Analyses Reveal Potential Multidrug Resistance Functions for SAV2435 and CTR107 and Other GyrI-like Proteins. *Biochemistry*

- 55, 4850-4863, (2016).
- 9 Newberry, K. J. *et al.* Structures of BmrR-drug complexes reveal a rigid multidrug binding pocket and transcription activation through tyrosine expulsion. *J. Biol. Chem.* **283**, (2008).
- 10 Gupta, K., Liao, J., Petrova, O. E., Cherny, K. E. & Sauer, K. Elevated levels of the second messenger c-di-GMP contribute to antimicrobial resistance of *Pseudomonas aeruginosa*. *Mol. Microbiol.* **92**, 488-506, (2014).
- 11 Holm, L. & Rosenstrom, P. Dali server: conservation mapping in 3D. *Nucleic Acids Res.* **38**, W545-549, (2010).
- 12 Das, T. & Manefield, M. Pyocyanin Promotes Extracellular DNA Release in *Pseudomonas aeruginosa*. *Plos One* **7**, (2012).
- 13 Das, T., Kutty, S. K., Kumar, N. & Manefield, M. Pyocyanin Facilitates Extracellular DNA Binding to *Pseudomonas aeruginosa* Influencing Cell Surface Properties and Aggregation. *Plos One* **8**, (2013).
- 14 Whitchurch, C. B., Tolker-Nielsen, T., Ragas, P. C. & Mattick, J. S. Extracellular DNA required for bacterial biofilm formation. *Science* **295**, 1487-1487, (2002).
- 15 Olsen, I. Biofilm-specific antibiotic tolerance and resistance. *Eur. J. Clin. Microbiol.* **34**, 877-886, (2015).
- 16 D'Alvise, P. W., Magdenoska, O., Melchiorson, J., Nielsen, K. F. & Gram, L. Biofilm formation and antibiotic production in *Ruegeria mobilis* are influenced by intracellular concentrations of cyclic dimeric guanosinmonophosphate. *Environ Microbiol.* **16**, 1252-1266, (2014).

Reviewers' comments:

Reviewer #1 (Remarks to the Author):

The manuscript "FBrIR from *Pseudomonas aeruginosa* is a common receptor for both cyclic di-GMP and pyocyanin" by Wang et al. has been revised in response to the reviewer comments. However, while the authors addressed various concerns, not all concerns were adequately addressed. Some of the remaining concerns are listed below. Also, while this was not indicated by any of the reviewer, it seems that the authors are unaware that the structure of BrIR has already been published. The structure was published in *Biochemical and Biophysical Research Communications* in August 2017 by Raju and Sharma. The article was already available in June 2017, ahead of print before the publication date. It would have been interesting for the authors to compare their structure with that obtained by Raju and Sharma.

1. The c-di-GMP and pyocyanin binding studies have improved, with the KD value for c-di-GMP and even pyocyanin now being more realistic. The authors now indicate the Kd for c-di-GMP to be 7.33.
2. Based on competition data, the authors were able to confirm a 1:1 ratio of c-di-GMP to BrIR. However, I am unclear how this ratio was obtained given that the data in supplemental Fig. 2 show that a total of 160 μ M c-di-GMP is needed to outcompete all F-c-di-GMP bound to 320 μ M of BrIR. This appears to be more like a 1:2 stoichiometry.
3. Previous reviewers asked for quantitative densitometry data of all EMSAs. Only two have been provided – but there are 10 EMSAs altogether. So the authors still refer to binding differences in terms of relative binding, see "Both Y183A and Y249A exhibited 35% a slightly weaker response to pyocyanin binding"
4. Reviewers asked to include EMSAs for BrIR variants harboring only q substitution in the c-di-GMP binding site (original data included EMSAs for BrIR harboring up to 4 substitutions). These data were not provided. Such EMSAs would have been helpful demonstrating that all 4 sites were equally important for c-di-GMP binding.
5. Relevance of c-di-AMP binding is unclear.
6. The authors should also provide an explanation why differing concentrations of BrIR and DNA were used. For example, Figure 4 indicates EMSAs being done with 6 or 15 nmol while Figure 6 shows EMSAs carried out using 6 and 24 nmol. Likewise, the concentration of the DNA probe varied. Why does the concentration of BrIR vary between ' μ M' and 'nmol', see Fig. 4 and 6??
7. Binding of other compounds (see antibiotics) still appears to be at too high concentrations, with some antibiotics not reaching saturation at concentrations above 10 μ M.
8. Of additional concern is the oligomerization state of BrIR. While the authors now included a calibration for the gel filtration study, the calibration curve is not shown in [ml], while the elution profile is. Moreover, BrIR and BmrR elute at almost identical elution volumes, but the Mr of both proteins is supposedly markedly different. Additionally, the oligomerization state after crosslinking does not fit with BrIR being a tetramer, given that even after 30 min of cross-linking, the majority of BrIR appears to be a dimer.
9. What is a dimer? I assume the authors refer to a dimer.

10. Of additional concern are the data based on the BrIR-C domain including the EB binding and competition between c-di-GMP and EB. Previous findings indicated truncated BrIR has no activity in vivo, not even when both fragments were provided via separate plasmids, regardless of the growth conditions. However, here, the authors demonstrate truncated BrIR to bind diverse compounds including c-di-GMP. While I understand that the authors are trying to make a point that BrIR, truncated or not, binds diverse molecules, the findings raise the question whether the binding has any meaning.

11. What is "FBrIR", see title.

Reviewer #2 (Remarks to the Author):

This revised manuscript by Wang *et al*. does much to address the points raised in the critique of this reviewer as well as Reviewer 1. However, there are still a few issues that must be addressed or corrected.

The biggest issue to address concerns the newly added studies that show BrIR binds c-di-AMP. Indeed, the authors show reasonable binding of this compound, which as noted is not found in *Pseudomonas aeruginosa*. However, their model of c-di-AMP binding to BrIR cannot be correct. Indeed, they have the side chain of an arginine interacting with N6 of adenine in one site and two arginine side chains interacting with the N6 of the second site. N6 and the nitrogens of the arginine side chain are both hydrogen bond donors. Hence, these contacts are invalid and would be especially problematic for c-di-AMP binding to C2E2. Further, the authors would appear to think that c-di-AMP binding to BrIR has the same stoichiometry as c-di-GMP binding to BrIR. Given the unlikely binding of c-di-AMP at C2E2, it is more likely that the stoichiometry is only 1 c-di-AMP bound per BrIR subunit. The authors should determine the stoichiometry of the BrIR-(c-di-AMP) complex. If the stoichiometry is equal to 2 c-di-AMPs per subunit, then the binding mode must be different at C2E2 or this signalling molecule is bound somewhere else. Also, the binding of c-di-AMP at C2E1 must be somewhat different as well because of the N6-guanidinium side chain proximity. Hydrogen bond donors cannot be next to hydrogen bond donors. This clearly has an impact on their statement/conclusion on page 13, lines 378-380, which now must be restated.

The authors appear to include too many significant digits for their binding constants and standard deviations. Please modify these.

Figure 4, panels f & g: The authors should include the CD spectrum of their BrIR-C2Emut3, especially given all the changes in this single protein and its strong phenotype. Is this protein as stable as the others?

The authors do not address properly a point raised in the previous critique. It is now on page 10, lines 291-292: "The BrIR-C domain-pyocyanin crystals do not diffract poorly because of the low solubility of pyocyanin." Low solubility of pyocyanin is not the reason for poor diffraction. Their crystals diffract poorly due to poor crystal packing or internal disorder. There are myriad reasons for the poor crystal packing and low solubility of an added ligand is not one of them. The authors should simply state that despite great effort, they were unable to obtain diffraction-quality crystals of the BrIR-pyocyanin complex.

General Comment:

Overall the manuscript is fairly well written but there are still several issues with poorer grammar and word choice. The authors must take extra care to fix these areas of poor English.

Response to the reviewers' comments:

Reviewer #1 (Remarks to the Author):

The manuscript “FBrIR from *Pseudomonas aeruginosa* is a common receptor for both cyclic di-GMP and pyocyanin” by Wang et al. has been revised in response to the reviewer comments. However, while the authors addressed various concerns, not all concerns were adequately addressed. Some of the remaining concerns are listed below. Also, while this was not indicated by any of the reviewer, it seems that the authors are unaware that the structure of BrIR has already been published. The structure was published in *Biochemical and Biophysical Research Communications* in August 2017 by Raju and Sharma. The article was already available in June 2017, ahead of print before the publication date. It would have been interesting for the authors to compare their structure with that obtained by Raju and Sharma.

AU: We have included the comparison of our BrIR structures and the recently reported BrIR-c-di-GMP structure by Raju and Sharma¹ (Supplementary Fig. 2). It turns out that Raju and Sharma's structure is very similar to ours and also suggests a 1:2 ratio for BrIR to c-di-GMP. The difference is we determined three BrIR related structures and included many *in vitro* and *in vivo* data to explain the molecular mechanism in detail. With all these data we confirmed that BrIR is a common receptor for both c-di-GMP and pyocyanin. We thank the reviewer for reminding us of the newly published paper about BrIR structure. Our manuscript was submitted to *Nature Communication* on May 30th of 2017. According to the BBRC's record, the paper authored by Raju and Sharma was submitted to the journal on Jun 8th of 2017, and accepted and available on-line on Jun 9th of 2017. This is why their work was not cited in our previous version of manuscript, and to be honest, through reading the published scientific paper regarding *P. aeruginosa* when we wrote the manuscript, we were not aware that they share the same interest with us in BrIR structure. We are very happy that their structure provides additional evidence for our findings.

1. The c-di-GMP and pyocyanin binding studies have improved, with the KD value for c-di-GMP and even pyocyanin now being more realistic. The authors now indicate the Kd for c-di-GMP to be 7.33.

AU: Thanks. The reviewers' comments and suggestions for our previous version of manuscript really helped us to improve the work.

2. Based on competition data, the authors were able to confirm a 1:1 ratio of c-di-GMP to BrIR. However, I am unclear how this ratio was obtained given that the data in supplemental Fig. 2 show that a total of 160 μ M c-di-GMP is needed to outcompete all F-c-di-GMP bound to 320 μ M of BrIR. This appears to be more like a 1:2 stoichiometry.

AU: We appreciate the reviewer's carefulness. To determine the binding stoichiometry of c-di-GMP to BrIR, 1 nM F-c-di-GMP was mixed with 160 μ M c-di-GMP and incubated with increasing amounts of BrIR. The resulting data exhibited a linear increase in the observed millipolarization until the saturation of c-di-GMP by BrIR (88 μ M). Therefore, the ratio of c-di-GMP to BrIR is 2:1, which is consistent with our structural data (Fig. 2a,h). The '160 μ M' in the previous figure legend of the supplemental Fig. 2 is a typo, which is corrected to 760 μ M in

the current Supplementary Fig 4.

3. Previous reviewers asked for quantitative densitometry data of all EMSAs. Only two have been provided – but there are 10 EMSAs altogether. So the authors still refer to binding differences in terms of relative binding, see “Both Y183A and Y249A exhibited a slightly weaker response to pyocyanin binding”

AU: All EMSAs were quantified by band densitometry as suggested, and the results support our claims in the manuscript. Some figures have been put in the supplementary material due to the word limitation of the journal (Supplementary Fig. 9).

4. Reviewers asked to include EMSAs for BrlR variants harboring only q substitution in the c-di-GMP binding site (original data included EMSAs for BrlR harboring up to 4 substitutions). These data were not provided. Such EMSAs would have been helpful demonstrating that all 4 sites were equally important for c-di-GMP binding.

AU: As suggested, we provide EMSAs results for all six single-point mutations (R31A, D35A, Y40A, R67A, R86A, and Y270A) of BrlR in the absence and presence of c-di-GMP. As shown in the Fig. 4d and 4e, all the single-point mutations exhibit a lower DNA binding ability than that of wild-type BrlR. Notably, the binding ability of R31A, Y40A, and R86A mutants significantly decreased compared to that of Wt BrlR. D35A and R67A showed a slight decrease in DNA-binding (Fig. 4d,e and Supplementary Note 2). BrlR WT binds ~5.5 folds more DNA in the presence of c-di-GMP than in the absence of c-di-GMP. For R31A, Y40A, Y270A, and R86A, the fold numbers are approximate 1.3, 1.1, 3.4, and 1.5, respectively, while the fold numbers for D35A and R67A are ~4.8 and ~5.2, respectively (Fig. 4d,e and Supplementary Note 2). In order to further investigate whether these amino acid residues of two separate c-di-GMP binding sites were equally important for the c-di-GMP binding, we conducted the FP binding experiments. The results (Supplementary Fig. 7a) show that the K_D values of R31A, Y40A, and Y270A variants are similar to the K_D value of C2E mut1 (R31A/D35A/Y40A/Y270A). The mutant D35A is the only one that shows a lower K_D value than that of C2E mut1 (Fig. 2g; Supplementary Fig. 7a; and Supplementary Note 1). These results indicate that R31, Y40 and Y270 play decisive roles in the first c-di-GMP binding site. As to the second c-di-GMP binding site, the R67A mutant has a K_D value lower than the K_D of R67A/R86A, and the R86A mutant has a K_D value similar to R67A/R86A (Fig. 2g; Supplementary Fig. 7a; and Supplementary Note 1), which indicates that R86 plays a decisive role in the second c-di-GMP binding site. The EMSAs results are consistent with the results for c-di-GMP binding by the FP assay. Meaning while, the circular-dichroism (CD) spectroscopy assays show that all six site-directed mutations do not cause major change in the secondary structure of BrlR (Supplementary Fig. 7b).

5. Relevance of c-di-AMP binding is unclear.

AU: The section for BrlR and c-di-AMP binding has been rewritten. In the current revision, we first determined the binding stoichiometry of c-di-AMP to BrlR by using the competitive FP binding experiment (Supplementary Fig. 5), and the result reveals a 1:1 ratio for BrlR:c-di-AMP binding. Second, the binding of BrlR-C2Emut1 and BrlR-C2Emut2 to c-di-AMP was analyzed by the FP assay. BrlR-C2Emut1 is defective in c-di-AMP binding with a K_D of $63.7 \pm 5.1 \mu\text{M}$ (a five-fold reduction compared to that of wild-type BrlR), while BrlR-C2Emut2 has a K_D of $10.3 \pm$

0.2 μM for c-di-AMP binding (similar with that of wild-type BrlR) (Fig. 2i and Supplementary Note 1). These results show that c-di-AMP bind to the first c-di-GMP binding site but not to the second c-di-GMP binding site of BrlR. Moreover, we demonstrate the differences in the binding mode of c-di-GMP and c-di-AMP more clearly in the revision.

6. The authors should also provide an explanation why differing concentrations of BrlR and DNA were used. For example, Figure 4 indicates EMSAs being done with 6 or 15 nmol while Figure 6 shows EMSAs carried out using 6 and 24 nmol. Likewise, the concentration of the DNA probe varied. Why does the concentration of BrlR vary between ' μM ' and 'nmol', see Fig. 4 and 6??

AU: In this revision, we have unified the concentrations of the DNA probes and protein in EMSAs. Overall, the binding of BrlR to DNA is weak. Since the binding of BrlR to *PmexA* and *PmexE* are much weaker than that to *PbrlR*, the concentration of BrlR was increased from 15 μM to 24 μM in the assay for BrlR binding to *PmexA* and *PmexE* (Fig. 6b,c). New EMSAs figures (Fig. 4c, Fig. 6a, and Fig. 6f) have been obtained, in which the concentration of c-di-GMP is reduced according to the new K_D value.

Interestingly, when we optimized the concentrations of c-di-GMP and pyocyanin in the EMSAs assay for this revision, a synergistic effect was observed. The binding affinity of BrlR to *PbrlR* is enhanced by the co-presence of c-di-GMP and pyocyanin at a relatively low concentration (Fig. 7b). We are curious about why this synergistic effect on BrlR-DNA binding was not observed when the high concentration of c-di-GMP (3 mM) was used with pyocyanin (100 μM) in the previous assays. By looking into the literatures, we found a report that c-di-GMP forms G-quadruplex structures at high concentration, thus the pyocyanin-shaped molecules end-stack or intercalate into these supramolecular polymer structures², which might interfere with the direct interaction between pyocyanin and BrlR.

7. Binding of other compounds (see antibiotics) still appears to be at too high concentrations, with some antibiotics not reaching saturation at concentrations above 10 mM.

AU: We totally agree with this reviewer. The observation of high-concentration dependent binding of antibiotics to BrlR actually provided us an important clue that BrlR might have a specific ligand rather than these antibiotics. It has been reported that BmrR, a BrlR homologous protein, binds diverse antibiotics in its C-terminal multidrug binding domain³. BrlR also has a conserved C-terminal multidrug-binding domain similar to that in BmrR, which suggests that BrlR may also recognize a variety of drug like molecules. In PAO1 strain, the overexpression of BrlR confers the strain resistance to tobramycin^{4,5}, therefore, we determined the K_D values for BrlR binding to tobramycin and other antibiotics. The binding affinity of BrlR for tobramycin is low and in the millimolar (mM) range, which indicates that BrlR might not be the direct sensor of tobramycin. This result prompted us to find a real specific ligand for BrlR, thus we demonstrated that pyocyanin is the one. Most of this section is in Supplementary Note 3 of current revision.

8. Of additional concern is the oligomerization state of BrlR. While the authors now included a calibration for the gel filtration study, the calibration curve is not shown in [ml], while the elution profile is. Moreover, BrlR and BmrR elute at almost identical elution volumes, but the M_r of both proteins is supposedly markedly different. Additionally, the oligomerization state after crosslinking does not fit with BrlR being a tetramer, given that even after 30 min of cross-linking,

the majority of BrIR appears to be a dimer.

AU: In the current revision, the calibration curve of size exclusion chromatography is shown in [ml] as suggested. In addition, we added the data of analytical ultracentrifugation, and the results of both assays show that BrIR is a tetramer in solution (Fig. 1f and Supplementary Fig. 1). As to the crosslinking experiment of BrIR, we used the same crosslinker described in the assay from Sauer's group, and our results are similar with their published data⁵, which is not simply fit with BrIR being a tetramer, just as what this reviewer pointed out. However, the inconsistency of these results can be well explained by our observation from the crystal structure that BrIR is a dimer of dimers. Although BrIR is a tetramer, the four protomers are not identical. In our structure, BrIR is organized into two dimers (AD and BC). The interactions between the two protomers within a dimer, i.e. A and D in dimer AD or B and C in dimer BC, are stronger than the interactions between the two protomers from different dimers. Consequently, the two protomers within a dimer could be cross-linked for the first place (Fig. 1a), and the formation of crosslinking within the dimers might provide stereochemical hindrances for the further crosslinking between dimers. Therefore, we always observe a result of BrIR being dimer in the crosslinking assay although BrIR is tetramer in solution.

9. What is a dimer? I assume the authors refer to a dimer.

AU: Corrected.

10. Of additional concern are the data based on the BrIR-C domain including the EB binding and competition between c-di-GMP and EB. Previous findings indicated truncated BrIR has no activity in vivo, not even when both fragments were provided via separate plasmids, regardless of the growth conditions. However, here, the authors demonstrate truncated BrIR to bind diverse compounds including c-di-GMP. While I understand that the authors are trying to make a point that BrIR, truncated or not, binds diverse molecules, the findings raise the question whether the binding has any meaning.

AU: We think there are some misunderstandings about this issue. Our structures show that BrIR-C cannot bind c-di-GMP (Fig. 1d), and BrIR-C is a conserved GyrI-like domain that could bind diverse molecules such as EB and RH6G⁶. We performed FP assay to determine the K_D values for BrIR binding to c-di-GMP by using a fluoresceinated c-di-GMP analog (F-c-di-GMP). We also noticed that the fluorophore of F-c-di-GMP is structurally similar to the backbone of RH6G (Fig. 2f, Fig. 5b). This means that BrIR-C may bind the fluorophore of F-c-di-GMP, thus may interfere with the assay for the binding between BrIR and c-di-GMP. We measured the K_D value for BrIR-C binding with F-c-di-GMP ($88.5 \pm 9.9 \mu\text{M}$, Fig. 5d), which is much larger than that for BrIR binding F-c-di-GMP ($7.3 \pm 0.5 \mu\text{M}$, Fig. 2g). Hence, it indicates that the interference is not significant and our K_D value for BrIR binds c-di-GMP is credible. We also performed the competition experiments by using FP assays with BrIR-C and the fluorophore (F-c-di-GMP) to test whether tobramycin, gentamicin, and ethidium compete for the same binding site in BrIR-C. We totally agree that the function of BrIR depends on the intact protein which includes both the N-terminal HTH domain and the C-terminal GyrI-like domain. Although the binding may have no physical meanings, these assays verified the binding affinity of full-length BrIR for c-di-GMP, and more importantly, led to the finding of pyocyanin as the specific ligand for BrIR (Supplementary Fig. 10 and Supplementary Note 2).

11. What is “FBrlR”, see title.

AU: This typo has been corrected.

Reviewer #2 (Remarks to the Author):

This revised manuscript by Wang *et al.* does much to address the points raised in the critique of this reviewer as well as Reviewer 1. However, there are still a few issues that must be addressed or corrected. The biggest issue to address concerns the newly added studies that show BrlR binds c-di-AMP. Indeed, the authors show reasonable binding of this compound, which as noted is not found in *Pseudomonas aeruginosa*. However, their model of c-di-AMP binding to BrlR cannot be correct. Indeed, they have the side chain of an arginine interacting with N6 of adenine in one site and two arginine side chains interacting with the N6 of the second site. N6 and the nitrogens of the arginine side chain are both hydrogen bond donors. Hence, these contacts are invalid and would be especially problematic for c-di-AMP binding to C2E2. Further, the authors would appear to think that c-di-AMP binding to BrlR has the same stoichiometry as c-di-GMP binding to BrlR. Given the unlikely binding of c-di-AMP at C2E2, it is more likely that the stoichiometry is only 1 c-di-AMP bound per BrlR subunit. The authors should determine the stoichiometry of the BrlR-(c-di-AMP) complex. If the stoichiometry is equal to 2 c-di-AMPs per subunit, then the binding mode must be different at C2E2 or this signalling molecule is bound somewhere else. Also, the binding of c-di-AMP at C2E1 must be somewhat different as well because of the N6-guanidinium side chain proximity. Hydrogen bond donors cannot be next to hydrogen bond donors. This clearly has an impact on their statement/conclusion on page 13, lines 378-380, which now must be restated.

AU: We really appreciate the helpful comments, and totally agree with this reviewer. According to the suggestion, we determined the binding stoichiometry of c-di-AMP to BrlR using the FP assay, which reveals a 1:1 ratio of BrlR:c-di-AMP binding (Supplementary Fig. 5) and suggests that there is only one c-di-AMP binding site in BrlR. Indeed, BrlR with a mutation in C2E1 site is defective in c-di-AMP binding, revealing a K_D of $63.7 \pm 5.1 \mu\text{M}$ (a five-fold reduction than that of wild-type BrlR). However, BrlR with a mutation in C2E2 site has a K_D of $10.3 \pm 0.2 \mu\text{M}$ for c-di-AMP binding, which is very similar to that of wild-type BrlR (Fig. 2i and Supplementary Note 1). These results suggest that c-di-AMP bind to the first c-di-GMP binding site but not to the second c-di-GMP binding site of BrlR. In the revised manuscript, the differences between binding modes of c-di-GMP and c-di-AMP to BrlR are described (Supplementary Note 1).

The authors appear to include too many significant digits for their binding constants and standard deviations. Please modify these.

AU: As suggested, we have changed the significant digits for the K_D values.

Figure 4, panels f & g: The authors should include the CD spectrum of their BrlR-C2Emut3, especially given all the changes in this single protein and its strong phenotype. Is this protein as stable as the others?

AU: The CD spectra of BrlR-C2Emut3 and other BrlR mutants have been included in Figure 4f. The CD spectrum of BrlR-C2Emut3 is very similar to that of wild-type BrlR, although

BrIR-C2Emut3 has a lower solubility than wild-type BrIR. The low solubility makes it difficult to determine the K_D value for c-di-GMP binding of BrIR-C2Emut3.

The authors do not address properly a point raised in the previous critique. It is now on page 10, lines 291-292: “The BrIR-C domain-pyocyanin crystals do not diffract poorly because of the low solubility of pyocyanin.” Low solubility of pyocyanin is not the reason for poor diffraction. Their crystals diffract poorly due to poor crystal packing or internal disorder. There are myriad reasons for the poor crystal packing and low solubility of an added ligand is not one of them. The authors should simply state that despite great effort, they were unable to obtain diffraction-quality crystals of the BrIR-pyocyanin complex.

AU: So many thanks for the suggestion, and the related text was revised.

General Comment:

Overall the manuscript is fairly well written but there are still several issues with poorer grammar and word choice. The authors must take extra care to fix these areas of poor English.

AU: The writing was checked and revised carefully, and the manuscript was also reviewed by a native English speaker from Nature Research Editing Service (certificate 1867-49FC-489C-2645-2DAF).

Reference:

- 1 Raju, H. & Sharma, R. Crystal structure of BrIR with c-di-GMP. *Biochem Bioph Res Co* **490**, 260-264, doi:10.1016/j.bbrc.2017.06.033 (2017).
- 2 Nakayama, S., Zhou, J., Zheng, Y., Szmecinski, H. & Sintim, H. O. Supramolecular polymer formation by cyclic dinucleotides and intercalators affects dinucleotide enzymatic processing. *Future Sci OA* **2**, FSO93, doi:10.4155/fso.15.93 (2016).
- 3 Bachas, S., Eginton, C., Gunio, D. & Wade, H. Structural contributions to multidrug recognition in the multidrug resistance (MDR) gene regulator, BmrR. *Proc Natl Acad Sci U S A* **108**, 11046-11051, doi:10.1073/pnas.1104850108 (2011).
- 4 Liao, J. & Sauer, K. The MerR-like transcriptional regulator BrIR contributes to *Pseudomonas aeruginosa* biofilm tolerance. *J Bacteriol* **194**, 4823-4836, doi:10.1128/JB.00765-12 (2012).
- 5 Chambers, J. R., Liao, J., Schurr, M. J. & Sauer, K. BrIR from *Pseudomonas aeruginosa* is a c-di-GMP-responsive transcription factor. *Mol Microbiol* **92**, 471-487, doi:10.1111/mmi.12562 (2014).
- 6 Moreno, A. *et al.* Solution Binding and Structural Analyses Reveal Potential Multidrug Resistance Functions for SAV2435 and CTR107 and Other Gyrl-like Proteins. *Biochemistry-US* **55**, 4850-4863, doi:10.1021/acs.biochem.6b00651 (2016).

Reviewers' comments:

Reviewer #1 (Remarks to the Author):

Wang et al. have revised the manuscript "BrIR from *Pseudomonas aeruginosa* is a common receptor for both cyclic di-GMP and pyocyanin" in response to previous reviewer comments. Revisions include editing the manuscript for typos etc, clarifying many of the experiments and experimental data, and including reference to the already existing BrIR crystal structure, the manuscript has significantly improved. While I still have some concerns (explanation of why no tetramer is detected after cross-linking, detection of BrIR binding in the absence of c-di-GMP or other effectors, overstatement of there being a "dramatic" conformational change upon c-di-GMP binding, why does BrIR bind c-di-AMP in the first place, etc.), I believe the manuscript has improved significantly.

Reviewer #2 (Remarks to the Author):

This revised manuscript by Wang et al. addresses many of the major problems raised in the previous critiques. There are, however, still a few questions remaining and points to be addressed.

The more important point that the authors must address is concerned with the use of the fluoresceinated c-di-GMP and c-di-AMP molecules used in their binding assays and the results. This approach was first described in Tschowri *et al* (*Cell* 2014) and later by Schumacher *et al*. (*NAR*, 2017). As an aside the authors should also reference that later paper as it describes very interesting properties of c-di-GMP binding to a BldD. The authors should also state somewhere in their manuscript (page 12, line 349) that the structure of the PilZ-containing transcription factor, MrkH from *Pseudomonas aeruginosa*, has also been solved bound to c-di-GMP (Schumacher and Zeng, *PNAS*, 2016). More to the original point, the authors seen only a 5-fold reduction in c-di-GMP binding when testing the BrIR-C2E-mut1 protein, which is surprisingly small given that many of the c-di-GMP interacting residues have been mutated. Furthermore, when singly mutated, as shown in Supplementary Figure 7A, the costliest mutation is the R31A protein. However, this is still only in the range of 4 to 5 fold loss of binding affinity. Why is it that the combined mutations of residues observed in the structure that interact directly with the c-di-GMP do not have a greater impact on the binding affinity? Why is there no single "killer" mutation as found in the BldD-(c-di-GMP) complex as shown by Schumacher *et al*., *NAR* 2017)? The concern that these results raise is focussed on a statement that the authors make later in the paper which is found on lines 256-258. If the backbones of pyocyanin and fluorescein are structurally similar as stated, should not the authors be concerned that the their fluoresceinated c-di-GMP might be binding to the pyocyanin site as well as to their other observed c-di-GMP site? If there is binding it might mask the results of their binding studies to the mutants. Hence, the authors should consider a simple binding experiment of fluorescein to BrIR.

Figure 6 Legend, lines 834-835. The authors do not include the secondary structure elements in this figure.

Minor issues:

Figure 3c and 3d: The authors should point out these are stereoviews.

Figure 6 Legend, lines 834-835. The authors do not include the secondary structure elements in this figure.

Page 7, lines 188-189: The authors should provide a reference or references for this statement.

Page 10, lines 272-274: The references are incomplete here. With regard to simultaneous binding by

two drugs, the authors should read and refer to the paper by Schumacher *et al.*, *EMBO J*, 2004. And lines 276-277, the authors should refer to the paper by Schumacher *et al.*, *Science*, 2001, in which the multisite, multidrug binding pocket was first described.

Reviewers' comments:

Reviewer #1 (Remarks to the Author):

Wang et al. have revised the manuscript “BrIR from *Pseudomonas aeruginosa* is a common receptor for both cyclic di-GMP and pyocyanin” in response to previous reviewer comments. Revisions include editing the manuscript for typos etc, clarifying many of the experiments and experimental data, and including reference to the already existing BrIR crystal structure, the manuscript has significantly improved. While I still have some concerns (explanation of why no tetramer is detected after cross-linking, detection of BrIR binding in the absence of c-di-GMP or other effectors, overstatement of there being a “dramatic” conformational change upon c-di-GMP binding, why does BrIR bind c-di-AMP in the first place, etc.), I believe the manuscript has improved significantly.

Thanks for the positive comments! Below are the responses to the reviewer’s concerns:

Concern: ‘**explanation of why no tetramer is detected after cross-linking**’

Response: We have found that BrIR is a tetramer by the gel-filtration as well as the analytical ultracentrifugation assays (Fig. 1f and Supplementary Fig. 1). We did observe trace amount of tetramer after cross-linking. However, much more dimers than tetramers were detected (Supplementary Fig. 13a). The native gel analysis of the same reactions showed that cross-linked BrIR is still a single band like the wild type protein, indicating that cross-linked BrIR is still a homogeneous oligomer. The faster migration rate of the modified BrIR most probably results from the reduction of positive charges due to the modification of the amines (Supplementary Fig. 13b). We then performed a gel filtration assay to verify that most of the cross-linked BrIR is still a tetramer in solution (Supplementary Fig. 13c). The slightly bigger elution volume of the modified BrIR may result from its slightly bigger molecular weight and volume.

The inconsistency of the results from different experiments may result from the chemical nature of cross-linking reaction between protein BrIR and the dithiobis (succinimidyl propionate)-DSP cross-linker. DSP has an amine-reactive N-hydroxysuccinimide (NHS) ester at each end of an 8-carbon spacer arm. The spacer length of DSP is 12.0 Å. NHS esters react specifically with the primary amines of lysine and the N-terminal of each polypeptide¹. Structural analysis indicates that each BrIR protomer contains only six lysine residues and all these lysines are located on the HTH motif and the C-terminal multidrug-binding domain of BrIR. The long α -helix linker of BrIR which mainly mediates BrIR tetramerization contains no lysine (Supplementary Fig. 13d). Since all cross-linkings occur between primary amines, the distances and steric hindrances between each pair of primary amines will determine the reaction result.

To make things clear we first check the contribution by the lysine residues. We measured the cross-distance matrix between each pair of primary amines of lysine from protomer A to B,C and D in apo BrIR (Supplementary Table 1 for review). The shortest distance between protomer A and protomer B is 31.4 Å (the counterpart is 35.5 Å in the BrIR-c-di-GMP structure), the shortest distance between protomer A and protomer C 31.6 Å (the counterpart is 45.0 Å in the BrIR-c-di-GMP structure), and the shortest distance between protomer A and protomer D 20.1 Å (the counterpart is 18.3 Å in the BrIR-c-di-GMP structure) (Table 1 for review). Considering the

spacer length of DSP is 12.0 Å and the length of the side chain of lysine is 6.3 Å, the intersubunit cross-linking most likely happen between protomer A and protomer D rather than between A and B or A and C. Strikingly, c-di-GMP binding may further promote the cross-linking between A and D (Supplementary Fig. 13f). For the same reason, cross-linking can also happens between protomer B and protomer C rather than B and D (Table 1 for review). Therefore, BrIR dimer is the major product of DSP cross-linking if lysine is the only residue gets involved in cross-linking. We thus check if the N-terminal amine also gets involved in cross-linking. Structural analysis shows that for each N-terminal amine there are two lysine residues within 20 Å (K208B and K241B is close to N-terminal amine of A, K208A and K241A close to N-terminal amine of B, K208D and K241D close to C, K208C and K241C close to D). It looks like that the N-terminal amine could be cross-linked to these to lysine residues and if this happens the major product should be tetramer considering the contribution of other lysine residues. However, two factors negatively affect the contribution of the N-terminal amine. First, Y270 makes a steric hindrance which is of no advantage to the cross-linking reaction. Secondly, the distance between K208 and K241 which belong to the same peptide chain is less than 10 Å. There is no steric hindrance between these two lysine residues. This means the cross-linking reaction between these two residues will outcompete the reaction with the N-terminal amine. Thus only a little bit tetramers form after cross-linking.

To further clarify the contribution of the N-terminal amine we performed the cross-linking of the N-terminal His₆-tagged BrIR (Its N-terminal amine would be too far to be linked to K208 or K241). The result shows that the amount of dimer almost remains the same as that of untagged BrIR while the already small amount of tetramer is further reduced (Supplementary Fig. 13e). In summary, cross-linking primarily happens between K215A and K69D (or K215C and K69B) thus produces the major product dimer. The N-terminal amine may have a little chance to be linked to K208 or K241 and produces a little tetramer (Tetramer may also formed by random collision between two dimers). The inconsistency between cross-linking result and other experiments was also reported in case of UV-laser cross-linking of Glyceraldehyde-3-Phosphate Dehydrogenase (GAPDH). GAPDH is a ~146 kDa tetramer which is a classical dimer of dimers. In this case only dimers can be observed after UV-laser cross-linking². All these results just reflect the intrinsic dimer-of-dimer organization.

Table 1 for review:

Distance (Å)		Protomer A					
		K16	K69	K163	K208	K215	K241
Protomer B	K16	NA	NA	41.8	NA	43.4	32.3
	K69	NA	NA	45.9	31.4	36.4	31.4
	K163	37.1	47.4	56.9	80.3	64.8	77.4
	K208	37.8	34.1	82.7	99.7	82.4	98.1
	K215	NA	36.7	66.4	80.4	66.4	77.8
	K241	31.9	31.5	80.9	99.3	80.6	98.2
Protomer C	K16	67.5	55.9	NA	79.2	NA	76.8
	K69	56.0	47.5	NA	80.7	NA	77.5
	K163	80.4	73.8	76.0	55.6	53.3	61.2
	K208	79.0	80.6	55.5	31.6	32.8	39.9
	K215	72.5	70.9	53.3	32.8	33.2	36.3

	K241	76.6	77.4	60.6	39.9	36.2	48.4
Protomer D	K16	50.1	50.1	62.1	57.0	38.9	63.7
	K69	50.1	55.6	43.5	39.9	20.1	46.6
	K163	64.9	46.5	96.7	93.0	81.0	94.4
	K208	57.5	39.3	98.5	104.0	89.9	103.3
	K215	39.7	20.8	82.3	87.1	70.5	88.1
	K241	63.6	46.4	99.4	103.3	90.8	102.1
Distance (Å)		Protomer B					
		K16	K69	K163	K208	K215	K241
Protomer A	K16	NA	NA	37.1	37.8	NA	31.9
	K69	NA	NA	47.4	34.1	36.7	31.5
	K163	41.8	45.9	56.9	82.7	66.4	80.9
	K208	NA	31.4	80.3	99.7	80.4	99.3
	K215	43.4	36.4	64.8	82.4	66.4	80.6
	K241	32.3	31.4	77.4	98.1	77.8	98.2
Protomer C	K16	49.9	50.0	64.9	57.3	39.7	63.7
	K69	50.0	55.5	46.4	39.9	20.8	46.4
	K163	62.1	43.5	96.7	98.1	82.2	99.4
	K208	57.0	40.1	93.0	103.8	87.1	103.2
	K215	39.0	20.1	80.9	89.5	70.5	90.8
	K241	63.8	46.7	94.4	103.3	88.1	102.1
Protomer D	K16	67.5	56.0	80.4	79.0	72.5	76.6
	K69	55.9	47.5	73.8	80.6	70.9	77.4
	K163	NA	NA	76.0	55.5	53.3	60.6
	K208	79.2	80.7	55.6	31.6	32.8	39.9
	K215	NA	NA	53.3	32.8	33.2	36.2
	K241	76.8	77.5	61.2	39.9	36.3	48.4

Concern: ‘detection of BrIR binding in the absence of c-di-GMP or other effectors’

Response: First, our crystal structures show that BrIR is a dimer of dimers in the absence or presence of c-di-GMP or other effectors (Fig. 1f, Fig. 2a, and Supplementary Fig. 1). Members of the MerR family assemble into dimers to bind DNA and exert their regulatory function³. The dimer-of-dimer organization of BrIR suggests it may have two binding interfaces to interact with DNA (Fig. 1c and Supplementary Fig. 14a, b). Second, the molecular ratio of BrIR to PbrIR is relatively high (320 : 1 or more) in the EMSA, which indicates that the interaction between BrIR and PbrIR is very weak in the absence of c-di-GMP or other effectors (Fig. 4a). Third, proteins are not rigid bodies, and they undergo dynamical structural rearrangements constantly⁴. At any given moment, there is always a small fraction of BrIR adopts the conformation in favor of DNA binding even in the absence of c-di-GMP or other effectors. According to the thermodynamic laws, although c-di-GMP (or other effectors) binding does not change the oligomerization state of BrIR, it does stabilize most of the BrIR tetramers in the right conformation in favor of strong DNA binding. Finally, Sauer’s group also found that BrIR can bind to its own promoter without any

effectors⁵, when the binding region of BrlR was tested by EMSA assays (Figure 2C in *Mol Microbiol.* 2014 May; 92(3): 471–487. doi:10.1111/mmi.12562.).

Concern: ‘overstatement of there being a “dramatic” conformational change upon c-di-GMP binding’

Response: Thanks for pointing out this overstatement. We rephrased the text in the section of structure comparison of apo-BrlR and BrlR/c-di-GMP complex. Indeed, the oligomerization state of BrlR is the same with or without c-di-GMP (Fig. 1f and Supplementary Fig. 1). c-di-GMP binding mainly changes the surface charge distribution of the HTH motif (Fig. 3e). Considering the long coiled-coil linker of BrlR is flexible, binding of c-di-GMP to the HTH motif will bend the coiled-coil linker and trigger a conformational change in the adjacent subunit (Fig. 3b).

Concern: ‘why does BrlR bind c-di-AMP in the first place’

The stacked conformation of c-di-AMP is similar to the binding mode of c-di-GMP in some proteins, such as the human STING protein⁶. BrlR has two binding sites for c-di-GMP (Fig. 2a), when we tentatively place a c-di-AMP in the first binding site, the interactions between BrlR and the c-di-AMP largely remain the same as the interactions between BrlR and the c-di-GMP (Supplementary Fig. 6a). When we try to place a c-di-AMP in the second binding site, however, the position which was occupied by H-bond acceptor O6 of the guanine from c-di-GMP is now occupied by an amino group (N6 atom of the adenine, which is an H-bond donor). This not only disrupts two H-bonds but also creates a like charge repulsion with guanidyls of R67 and R86 (Supplementary Fig. 6b). Consequently, c-di-AMP can only bind to the first binding site but not the second binding site for c-di-GMP. The 1:1 ratio of BrlR:c-di-AMP binding has been confirmed by both FP assay (Supplementary Fig. 5), and the mutagenesis study (Fig. 2i and Supplementary Note 1).

Reviewer #2 (Remarks to the Author):

This revised manuscript by Wang et al. addresses many of the major problems raised in the previous critiques.

Response: Thanks for the positive comment.

The more important point that the authors must address is concerned with the use of the fluoresceinated c-di-GMP and c-di-AMP molecules used in their binding assays and the results. This approach was first described in Tschowri et al (Cell 2014) and later by Schumacher et al. (NAR, 2017). As an aside the authors should also reference that later paper as it describes very interesting properties of c-di-GMP binding to a BldD. The authors should also state somewhere in their manuscript (page 12, line 349) that the structure of the PilZ-containing transcription factor, MrkH from *Pseudomonas aeruginosa*, has also been solved bound to c-di-GMP (Schumacher and Zeng, PNAS, 2016). More to the original point, the authors seen only a 5-fold reduction in c-di-GMP binding when testing the BrlR-C2E-mut1 protein, which is surprisingly small given that many of the c-di-GMP interacting residues have been mutated. Furthermore, when singly mutated, as shown in Supplementary Figure 7A, the costliest mutation is the R31A protein. However, this is still only in the range of 4 to 5 fold loss of binding affinity. Why is it that the combined mutations

of residues observed in the structure that interact directly with the c-di-GMP do not have a greater impact on the binding affinity? Why is there no single "killer" mutation as found in the BldD-(c-di-GMP) complex as shown by Schumacher et al., NAR 2017)? The concern that these results raise is focussed on a statement that the authors make later in the paper which is found on lines 256-258. If the backbones of pyocyanin and fluorescein are structurally similar as stated, should not the authors be concerned that their fluoresceinated c-di-GMP might be binding to the pyocyanin site as well as to their other observed c-di-GMP site? If there is binding it might mask the results of their binding studies to the mutants. Hence, the authors should consider a simple binding experiment of fluorescein to BrlR.

Response: We thank the reviewer for these helpful comments. We have added these references in the revision, especially in the fluorescence polarization (FP)-based binding assays. We quite understand the reviewer's concern about the FP-assays because we also worried that modified c-di-GMP might produce some errors in the experiments. So we first tried to use Isothermal titration calorimetry (ITC) to determine the binding affinity of BrlR and c-di-GMP. However, the signal is too weak to give a feasible result. We then tried Surface plasmon resonance (SPR) and got the K_D values. But the K_D values are too big to have physiological significance. The reason has been analyzed in detail in the previous response. Next we considered microscale thermophoresis (MST). However, to perform MST, a fluorescent group should be cross-linked to the primary amines of lysine and the N-terminal of BrlR. We already knew from the structures that M1 is part of the first c-di-GMP binding site. Thus, the fluorophore attached to the N-terminus of BrlR may significantly affect the c-di-GMP binding. Moreover, the fluorophore may also interact with BrlR-C domain and cause protein aggregation. Finally, we turned to FP-based binding assays according to the reviewer's suggestion and got a reasonable K_D value for BrlR binding to c-di-GMP (Fig. 2g).

As the reviewer mentioned, the backbones of pyocyanin and fluorescein are structurally similar (Fig. 5b). We also worried that the K_D values we got with F-c-di-GMP and F-c-di-AMP just reflect the interactions between the fluorophore and the BrlR-C domain. Actually, we did find that BrlR binds fluorescein with a K_D of $33.6 \pm 1.6 \mu\text{M}$ (Fig. 5c). This means our worry is not superfluous. However, the ligands we used in the assays are not just the fluorophore but F-c-di-GMP and F-c-di-AMP which are of much bigger size (Fig. 2f). The nucleotides may cause some steric hindrance that may weaken the binding to BrlR-C domain. Indeed, the FP binding assays show that BrlR-C, which cannot bind c-di-GMP, binds F-c-di-GMP with a K_D of $88.5 \pm 9.9 \mu\text{M}$ (Supplementary Fig. 11a). This K_D could be roughly seen as a background in the FP assays. Because this value is much higher than that of BrlR binding to F-c-di-GMP ($K_D = 7.3 \pm 0.5 \mu\text{M}$; Fig. 2g) or F-c-di-AMP ($K_D = 12.9 \pm 0.7 \mu\text{M}$; Fig. 2i), we know that BrlR binds to c-di-GMP or c-di-AMP much tighter than to fluorescein. The K_D value we determined is mainly contributed by the interaction between BrlR and c-di-GMP, considering that only 1.0 nM F-c-di-GMP was used in FP binding assays. Even the interaction between the fluorophore and the BrlR-C domain may cause some errors, the errors would be a small value. The consistence between the structures and the biochemistry data also support this point.

In the case of F-c-di-GMP binding to BrlR mutant (BrlR-C2E-mut1, $K_D = 36.2 \pm 2.5 \mu\text{M}$, BrlR-C2E-mut2, $K_D = 21.6 \pm 1.6 \mu\text{M}$, Fig. 2g), the K_D values should be also mainly contributed by the interaction between BrlR and c-di-GMP. Of course, because these K_D values are closer to the background K_D value, the error should be larger in this case. As the reviewer pointed out this is

right the reason that we can only observed about 5-fold reduction in c-di-GMP binding for BrlR-C2E-mut1 and costliest mutation R31A. For the same reason we can also not get an ordinary “killer” mutation which does not bind to ligand at all. Because the interaction between the fluorophore and BrlR-C domain, we can never get a K_D value lower than the background. However, the K_D values still reflect the effect of the mutations and tell us that the first binding site has a higher affinity for c-di-GMP. By comparing K_D value for each point mutation we still can find out which point mutation has the biggest impact on c-di-GMP binding.

c-di-AMP only binds to the first c-di-GMP binding site (Supplementary Fig. 5 and Supplementary Note 1), so BrlR-C2E-mut1 is defective in F-c-di-AMP binding with a K_D of $63.7 \pm 5.1 \mu\text{M}$ (Fig. 2i). This K_D value in fact reflects the fluorophore mediated interaction between F-c-di-AMP and BrlR-C domain, thus could be seen as the background K_D . In this case BrlR-C2E-mut1 could be seen as a practical “killer” mutation. This K_D value is similar to the K_D of BrlR-C with F-c-di-GMP ($88.5 \pm 9.9 \mu\text{M}$, Supplementary Fig. 11a). As BrlR can totally bind three F-c-di-GMPs (two in the two separate c-di-GMP binding sites, one in the pyocyanin binding site mediated by the fluorophore of F-c-di-GMP), it is impossible to get an ordinary single “killer” mutation as in the $\text{BldD}_2\text{-(c-di-GMP)}_4$ complex shown by Schumacher’s group⁷. Even a practical single “killer” mutation is also unavailable. We pointed out the disadvantages of FP binding assay clearly in the revision. However, even with these disadvantages FP assay is still better than ITC, SPR and MST in term of giving useful data to help us understand the interaction between BrlR and its ligands.

Minor issues:

Figure 3c and 3d: The authors should point out these are stereoviews.

Response: As suggested, we made it clear that the Figure 3c and 3d are stereoviews in the legend.

Figure 6 Legend, lines 834-835. The authors do not include the secondary structure elements in this figure.

Response: We have rewritten the legend in Figure 6e according to the reviewer’s suggestion.

Page 7, lines 188-189: The authors should provide a reference or references for this statement.

Response: As suggested, we have provided the reference.

Page 10, lines 272-274: The references are incomplete here. With regard to simultaneous binding by two drugs, the authors should read and refer to the paper by Schumacher et al., EMBO J, 2004. And lines 276-277, the authors should refer to the paper by Schumacher et al. Science, 2001, in which the multisite, multidrug binding pocket was first described.

Response: As suggested, we have added these references in this revision.

Reference:

- 1 Mattson, G. *et al. Mol. Biol. Rep.* **17**, 167-183 (1993).
- 2 Itri, F. *et al. Femtosecond UV-laser pulses to unveil protein-protein interactions in living cells. Cell Mol. Life Sci.* **73**, 637-648, doi:10.1007/s00018-015-2015-y (2016).

- 3 Kumaraswami, M., Newberry, K. J. & Brennan, R. G. Conformational Plasticity of the
Coiled-Coil Domain of BmrR Is Required for bmr Operator Binding: The Structure of
Unliganded BmrR. *J. Mol. Biol.* **398**, 264-275, doi:10.1016/j.jmb.2010.03.011 (2010).
- 4 Frauenfelder, H., Sligar, S. G. & Wolynes, P. G. The Energy Landscapes and Motions of
Proteins. *Science* **254**, 1598-1603, doi:DOI 10.1126/science.1749933 (1991).
- 5 Chambers, J. R., Liao, J., Schurr, M. J. & Sauer, K. BrlR from *Pseudomonas aeruginosa* is
a c-di-GMP-responsive transcription factor. *Mol. Microbiol.* **92**, 471-487,
doi:10.1111/mmi.12562 (2014).
- 6 Kranzusch, P. J. *et al.* Ancient Origin of cGAS-STING Reveals Mechanism of Universal
2',3' cGAMP Signaling. *Mol. cell* **59**, 891-903, doi:10.1016/j.molcel.2015.07.022 (2015).
- 7 Schumacher, M. A. *et al.* The *Streptomyces* master regulator BldD binds c-di-GMP
sequentially to create a functional BldD2-(c-di-GMP)₄ complex. *Nucleic Acids Res.* **45**,
6923-6933, doi:10.1093/nar/gkx287 (2017).

REVIEWERS' COMMENTS:

Reviewer #2 (Remarks to the Author):

This revised manuscript by Wang *et al.* addresses the remaining major problems raised in the previous critiques. There are, however, still an issue or two that the authors should address.

Page 16, lines 458-462: The authors should explain why the binding curves from their SPR experiments were fit using a 1:1 (c-di-GMP:BrIR subunit) binding model. Why is this not a 2:1 fit? Was a 2:1 fitting experiment tried?

With regard to the point raised by Reviewer 1 "Why does BrIR bind c-di-AMP in the first place?", it does not appear that the authors addressed this correctly. The authors should emphasise that *Pseudomonas aeruginosa* does not make c-di-AMP, but other bacteria with a closely related BrIR homologue do and hence, the authors are simply using the Pa BrIR as a model system to understand how c-di-AMP might bind to these other proteins. Is this the rationale behind their c-di-AMP binding studies?

Table 1: The authors appear to have too many significant digits in some of their values. They should fix these.

Table 1: "I3C dreivative" should be "I3C derivative"

Supplementary Figure 12a: Are the units of concentration shown in this figure correct? Are these micromolar or millimolar?

Supplementary Figure 5: Is this TipA from *Staphylococcus aureus* or from *Bacillus subtilis*?

Page 26, lines 575 and 579: "A close stereoview"; should be "A close-up stereoview"

The authors need to redo some of the Editorial Policy Checklists as some of the answers have changed, e.g., there are now stereoviews and this must be addressed and there appear to be no competing financial interests. Further, the data availability page reference is not correct.

Response to the reviewers' comments:

Reviewer #2 (Remarks to the Author):

This revised manuscript by Wang *et al.* addresses the remaining major problems raised in the previous critiques. There are, however, still an issue or two that the authors should address.

Page 16, lines 458-462: The authors should explain why the binding curves from their SPR experiments were fit using a 1:1 (c-di-GMP:BrIR subunit) binding model. Why is this not a 2:1 fit? Was a 2:1 fitting experiment tried?

AU: In the current version of manuscript, the SPR experiments are only employed to determine the interaction between BrIR and diverse toxic compounds, i.e. pyocyanin and some antibiotics.

In the first submitted version of manuscript, we determined all the interactions between BrIR and c-di-GMP, pyocyanin as well as other compounds by using SPR. However, as pointing out by the reviewers, SPR experiments gave an unreasonably high K_D value for c-di-GMP binding. In the first revision, we have explained the possible reason for that observation. Briefly, in SPR experiments, BrIR was immobilized on an activated CM5 sensor chip by amine coupling (BIAcore). Each BrIR tetramer has four terminal amines and thus could be immobilized by one or two covalent bonds considering the steric hindrance (Fig. 1a). Consequently, the covalent modification of the N-termini will cause a steric hindrance which affects c-di-GMP binding to the binding sites near N-termini. Moreover, c-di-GMP forms heterogeneous oligomers at high concentration. This means a very low effective concentration even large amount of c-di-GMP exists. These factors combine to give a very weak artificial affinity and a high K_D value for c-di-GMP. In the revision, we solved this problem by using fluorescence polarization (FP)-based binding assay and found out that one BrIR subunit binds two c-di-GMPs (Fig. 2g,h).

However, the pyocyanin binding site on BrIR-C domain is not affected by immobilization. From the structure (Fig. 5a,f), we know that BrIR only has one binding site for pyocyanin or other toxic compounds, so we use a 1:1 (pyocyanin or antibiotic :BrIR subunit) binding model in the SPR experiments. We explained this in the current revision (page 16, lines 455-456).

With regard to the point raised by Reviewer 1 "Why does BrIR bind c-di-AMP in the first place?", it does not appear that the authors addressed this correctly. The authors should emphasise that *Pseudomonas aeruginosa* does not make c-di-AMP, but other bacteria with a closely related BrIR homologue do and hence, the authors are simply using the Pa BrIR as a model system to understand how c-di-AMP might bind to these other proteins. Is this the rationale behind their c-di-AMP binding studies?

AU: Thanks the reviewer for helping us to clearly explain our rationale, and the related statements are now provided in the first paragraph of the discussion (page 11, lines 306-308).

Table 1: The authors appear to have too many significant digits in some of their values. They should fix these.

AU: We have fixed the significant digits of the values in Table 1.

Table 1: "I3C dreivative" should be "I3C derivative"

AU: Corrected.

Supplementary Figure 12a: Are the units of concentration shown in this figure correct? Are these micromolar or millimolar?

AU: We thank the reviewer for pointing out this issue, and the figure has been rearranged in the current version of manuscript.

Supplementary Figure 5: Is this TipA from *Staphylococcus aureus* or from *Bacillus subtilis*?

AU: We double-checked the sequence in Supplementary Figure 5 and confirmed that TipA is from *Staphylococcus aureus*.

Page 26, lines 575 and 579: "A close stereoview"; should be "A close-up stereoview"

AU: Corrected.

The authors need to redo some of the Editorial Policy Checklists as some of the answers have changed, e.g., there are now stereoviews and this must be addressed and there appear to be no competing financial interests. Further, the data availability page reference is not correct.

AU: Thank you! We have carefully made the check follow the checklists, and the data availability page has been corrected.